# MBL-1/Muscleblind regulates neuronal differentiation and controls the splicing of a terminal selector in *Caenorhabditis elegans*

**Ho Ming Terence Lee**, **Hui Yuan Lim**, **Haoming He**, **Chun Yin Lau**, **Chaogu Zheng** *

School of Biological Sciences, The University of Hong Kong, Hong Kong SAR, China

* cgzheng@hku.hk

## Abstract

The muscleblind family of mRNA splicing regulators is conserved across species and regulates the development of muscles and the nervous system. However, how Muscleblind proteins regulate neuronal fate specification and neurite morphogenesis at the single-neuron level is not well understood. In this study, we found that the *C. elegans* Muscleblind/MBL-1 promotes axonal growth in the touch receptor neurons (TRNs) by regulating microtubule stability and polarity. Transcriptomic analysis identified dozens of MBL-1-controlled splicing events in genes related to neuronal differentiation or microtubule functions. Among the MBL-1 targets, the LIM-domain transcription factor *mec-3* is the terminal selector for the TRN fate and induces the expression of many TRN terminal differentiation genes. MBL-1 promotes the splicing of the *mec-3* long isoform, which is essential for TRN fate specification, and inhibits the short isoforms that have much weaker activities in activating downstream genes. MBL-1 promotes *mec-3* splicing through three "YGCU(U/G)Y" motifs located in or downstream of the included exon, which is similar to the mechanisms used by mammalian Muscleblind and suggests a deeply conserved context-dependency of the splicing regulation. Interestingly, the expression of *mbl-1* in the TRNs is dependent on the *mec-3* long isoform, indicating a positive feedback loop between the splicing regulator and the terminal selector. Finally, through a forward genetic screen, we found that MBL-1 promotes neurite growth partly by inhibiting the DLK-1/p38 MAPK pathway. In summary, our study provides mechanistic understanding of the role of Muscleblind in regulating cell fate specification and neuronal morphogenesis.

## Author summary

The coding region of a gene is organized in multiple exons which are separated by the non-coding introns. The joining of different exons in different mRNA isoforms enables one gene to code for more than one protein through alternative mRNA splicing, which plays an important role in the development of the nervous system. In this study, we report the function of an evolutionarily conserved splicing regulator, called Muscleblind, in regulating neuronal differentiation and morphogenesis in the nematode *Caenorhabditis*

**Data Availability Statement:** All data are in the manuscript and/or supporting information files.

**Funding:** This work is supported by grants from the National Natural Science Foundation of China

(Excellent Young Scientists Fund for Hong Kong and Macau 32122002 to C.Z.), the Research Grants Council of Hong Kong (GRF 17106322, GRF 17107021, GRF 17105523, and CRF C7026-20G to C.Z.), the Food and Health Bureau of Hong Kong (HMRF 09201426 to C.Z.), and the University of Hong Kong (seed fund 202011159053). The funders had no role in study design, data collection and analysis, decision to publish, or preparation of the manuscript.

**Competing interests:** The authors have declared that no competing interests exist.

*elegans*. We found that deleting the muscleblind gene *mbl-1* led to defects in axonal growth and instability of microtubules, an essential component of the cytoskeleton. Using a transcriptomic approach, we discovered multiple MBL-1 downstream genes, whose mRNA splicing pattern is altered in mutants that lacked MBL-1. One target gene is the transcription factor *mec-3*, which is required for the differentiation of a type of mechano-sensory neurons called touch receptor neurons. MBL-1 promotes the splicing of the long and active isoform of MEC-3 and inhibits the generation of the short and less active isoform of MEC-3. By switching the *mec-3* mRNA from short to long isoforms, MBL-1 promotes the activation of MEC-3-regulated genes and safeguards the differentiation of the touch receptor neurons.

## Introduction

Alternative mRNA splicing serves as a critical step in the post-transcriptional regulation of gene expression and significantly increases transcriptomic and proteomic diversity. A genome-wide comparison of the levels of alternative splicing across human tissues found that the brain has the highest fraction of alternatively spliced genes [1], suggesting the importance of mRNA splicing regulation in the nervous system. In fact, alternative splicing has been found to regulate neuronal fate specification, neuronal migration, cellular morphogenesis, and synaptic formation during neurodevelopment [2]. For example, the switch from the expression of RNA-binding protein PTBP1 to its neuronal paralog PTBP2 promotes the differentiation of neural progenitor cells into neurons. PTBP1 suppresses the splicing of some neural targets to inhibit neuronal differentiation, whereas PTBP2 activates them to promote differentiation [3]. PTBP2 is subsequently downregulated in postmitotic neurons during synaptic formation to allow functional expression of postsynaptic density protein-95 (PSD-95) via splicing control [4]. Glia cells and neurons often express distinct isoforms of the same gene (e.g., *pyruvate kinase M*) [5], and different neuron types can also express differentially spliced isoforms (e.g., *unc-16* in GABAergic and cholinergic motor neurons in *C. elegans*) [6]. RNA splicing regulator Nova2 suppresses the inclusion of exon 9b/c in *Disabled-1*, safeguarding neuronal migration in mice [7], while neuronal activity induces alternative splicing of neurexin-1 to modulate synaptic transmission [8].

The Muscleblind family of tissue-specific alternative splicing regulators is particularly interesting because of their evolutionarily conserved RNA recognition domain (the tandem CCCH zinc finger domain) and their involvement in type 1 myotonic dystrophy. Mutations in *Drosophila* Muscleblind (Mbl) lead to defects in photoreceptor differentiation and muscle development (hence its name) [9], whereas the human homolog MBNL proteins participate in the differentiation of muscle cells, neurons, adipocytes, and blood cells [10]. In myotonic dystrophy, the expansion of non-coding CUG or CCUG repeats in the DMPK gene sequesters functional MBNL, causing defects similar to MBNL loss-of-function phenotypes [11–13]. The lack of available MBNL results in the mis-splicing of many target genes, including the cardiac troponin T, insulin receptor, chloride channel CLCN1, etc. The loss of CLCN1 may lead to myotonia [14].

Compared to the well-characterized functions of Muscleblind in muscle cells, the activity of Muscleblind in the nervous system is less understood. Cytoplasmic MBNL1 promotes neurite morphogenesis in cultured hippocampal neurons [15], while the deletion of MBNL2 reduces synaptic transmission and impairs hippocampal synaptic plasticity [16]. Conditional knockout of both MBNL1 and MBNL2 in the mouse brain affected cortical neuron distribution, reduced

dendritic complexity, and altered postsynaptic density morphology [17]. In *Drosophila*, Mbl regulates *Dscam2* cell-specific alternative splicing and promotes axonal and dendritic growth [18]. In *C. elegans* motor neurons, Muscleblind homolog MBL-1 regulates synaptic formation at neuromuscular junctions [19]. Despite these previous studies, the mechanisms by which Muscleblind and alternative splicing regulate neurite outgrowth remain poorly understood. In particular, the target splicing events and downstream signaling regulated by Muscleblind during neuronal differentiation remain largely unknown.

In this study, we found that the *C. elegans* Muscleblind/MBL-1 promotes neurite growth and cargo transport by regulating microtubule (MT) stability and polarity in the mechanosensory touch receptor neurons (TRNs). Transcriptomic analysis identified dozens of MBL-1 target splicing events, many of which affected genes involved in neuronal differentiation or MT functions. Among the target genes, we found that MBL-1 regulated the splicing of the LIM-homeodomain transcription factor *mec-3* by promoting the generation of its long and active isoform, which in turn induced the expression of TRN differentiation genes, including tubulins and MT-related genes. MBL-1 regulated *mec-3* alternative splicing through classical "YGCU(U/G)Y" motifs. From a forward genetic screen, we also identified the DLK-1/p38 MAPK pathway to be downstream of MBL-1; inactivating the DLK-1 pathway partially rescued the neurite growth defects in *mbl-1(-)* mutants. Overall, our study highlights an evolutionarily conserved function of the RNA-binding protein Muscleblind in regulating neuronal fate specification and neurite morphogenesis.

## Results

### RNA splicing regulator MBL-1/Muscleblind promotes neurite growth

To identify regulators of MT stability and neurite growth, we previously conducted a genetic screen using the *mec-7(u278)* mutants as a sensitized background [20]. *mec-7* codes for a TRN-specific β-tubulin, and *u278* is a neomorphic allele that caused the formation of hyperstable MTs and the production of an ectopic, posteriorly directed neurite in the ALM neurons, called ALM-PN. ALM and PLM neurons are the two major subtypes of the TRNs in *C. elegans*. In the *mec-7(u278)* suppressor screen, we searched for mutants that led to the loss of the ectopic ALM-PN and identified a missense allele *u1178* in *mbl-1*, which codes for an RNA splicing regulator orthologous to Drosophila Mbl and human MBNL.

Like the *Drosophila* Mbl, *C. elegans* MBL-1 contains two C-x8-C-x5-C-x3-H (CCCH) type zinc finger domains in tandem at the N-terminus, which interacts with RNA. A conserved cysteine in the second zinc finger domain was mutated to tyrosine in the *u1178* (C86Y) allele, suggesting that the mutant protein may be defective in RNA binding. Both *u1178* and a deletion allele *tm1563*, which removed an exon shared by all *mbl-1* isoforms and caused frameshift (Fig 1E), completely suppressed the generation of ALM-PN in *mec-7(u278)* mutants (Fig 1A and 1C). The *tm1563* allele was used as *mbl-1(-)* in the following study. The loss of MBL-1 also suppressed the growth of ALM-PN induced by elevated MT stability in *klp-7(-)* mutants; *klp-7* codes for a MT-depolymerizing kinesin-13 (S1A Fig). Thus, the effect of *mbl-1(-)* is not specific to the *mec-7(u278)* genetic background. While we were conducting this study, Puri *et al*. reported the identification of *mbl-1* as a suppressor of the ALM-PN phenotype in the *klp-7(-)* mutants in an independent suppressor screen [21]. Our results confirmed their findings. Mutations in *mbl-1* also affected normal TRN development and caused the shortening of PLM-AN, which did not extend beyond the vulva in *mbl-1(-)* mutants (Fig 1B and 1D). These results suggested that MBL-1 promotes TRN neurite growth.

The *mbl-1* gene generates multiple isoforms with different transcription start sites (Fig 1E). To study *mbl-1* expression patterns, we cloned ~5 kb upstream sequences before the start of

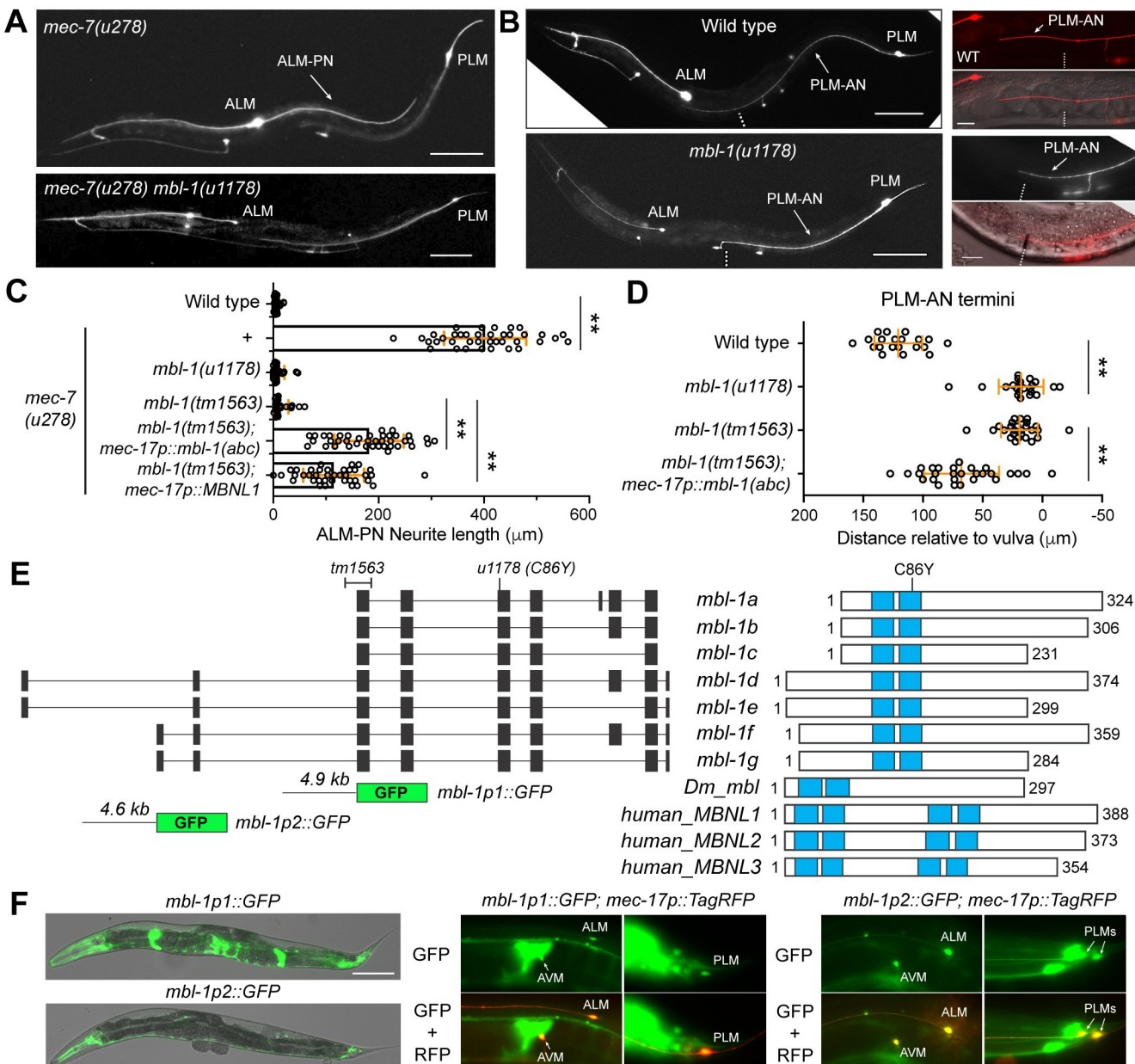

**Fig 1. Mutations in *mbl-1* suppressed neurite growth in the TRNs.** (A) A long ALM-PN in *mec-7(u278)* animals and the absence of ALM-PN in *mbl-1 (u1178; C86Y); mec-7(u278)* mutants isolated from the suppressor screen. The transgene *uIs115[mec-17p::TagRFP]* was used to visualize the TRNs. (B) Compared to the wild-type animals, *mbl-1(u1178)* animals had shortened PLM-AN that terminated at the position around the vulva (dashed line). The left panels showed 10x images; scale bar = 100 μm. The right panels showed 40x images; scale bar = 20 μm. (C) Quantification of ALM-PN length in various strains; *mec-17p::mbl-1(abc)* indicated the strain with TRN-specific expression of *mbl-1a-c* isoforms from the genomic region that encodes these isoforms; *mec-17p::MBNL1* indicated the expression of human *MBNL1* variant 21. Double asterisks indicate $p < 0.01$ in a post-ANOVA Tukey's HSD test. (D) Quantification of PLM-AN length by the distance from the vulva to the PLM-AN terminus. Positive values mean that PLM-AN grew beyond the vulva, while negative values mean PLM-AN failed to reach the vulva. (E) Exon structure of *mbl-1* isoforms and the domain structures of the proteins they code for. The blue blocks indicate zinc finger motifs. (F) GFP signal from the two promoter reporters for *mbl-1* (structure shown in E). The left panel are merged images of the GFP and DIC channel at 10x (scale bar = 100 μm). The middle and the right panels showed the overlapping with the TRN marker *uIs115[mec-17p:: TagRFP]* to show the expression of GFP in TRNs (scale bar = 20 μm).

*mbl-1a* and *mbl-1f* isoforms and generated *mbl-1p1*::*GFP* and *mbl-1p2*::*GFP* reporters, respectively. Both reporters produced GFP expression in the nervous system including the TRNs, PVDs, and the ventral nerve cord neurons (Fig 1F). Only *mbl-1p1*::*GFP* was strongly expressed in the spermatheca and the uterus. TRN expression of *mbl-1* was also confirmed using a fosmid-based translational reporter *wgIs664[mbl-1*::*EGFP]*, which showed that MBL-1 protein was located in the cell body with enrichment in the nucleus (S1B Fig). To confirm that *mbl-1* functions cell autonomously, we cloned the genomic fragment of *mbl-1(+)* that codes for the *a*, *b*, and *c* isoforms. Expression of these isoforms under the TRN-specific *mec-17* promoter rescued the ALM-PN growth in *mec-7(u278) mbl-1(-)* mutants and the PLM-AN growth in *mbl-1 (-)* mutants (Fig 1C and 1D). These results indicated that MBL-1 acted within the TRNs to regulate neurite growth. Moreover, the human ortholog MBNL1 also partially rescued the loss of *C. elegans mbl-1* (Fig 1C), suggesting that the function of the Muscleblind proteins is evolutionarily conserved.

## MBL-1/Muscleblind regulates microtubule stability

Next, we examined the effects of MBL-1 on MT stability. To monitor MT dynamics, we expressed a GFP-fused MT plus-end binding protein EBP-2 in the TRNs and visualized MT polymerization by tracking the movement of EBP-2::GFP. By counting the number of EBP-2:: GFP tracks in the PLM neurons, we found that mutations in *mbl-1* increased MT dynamics in both wild-type and *mec-7(u278)* backgrounds (Fig 2A and 2B). Since MTs in the wild-type

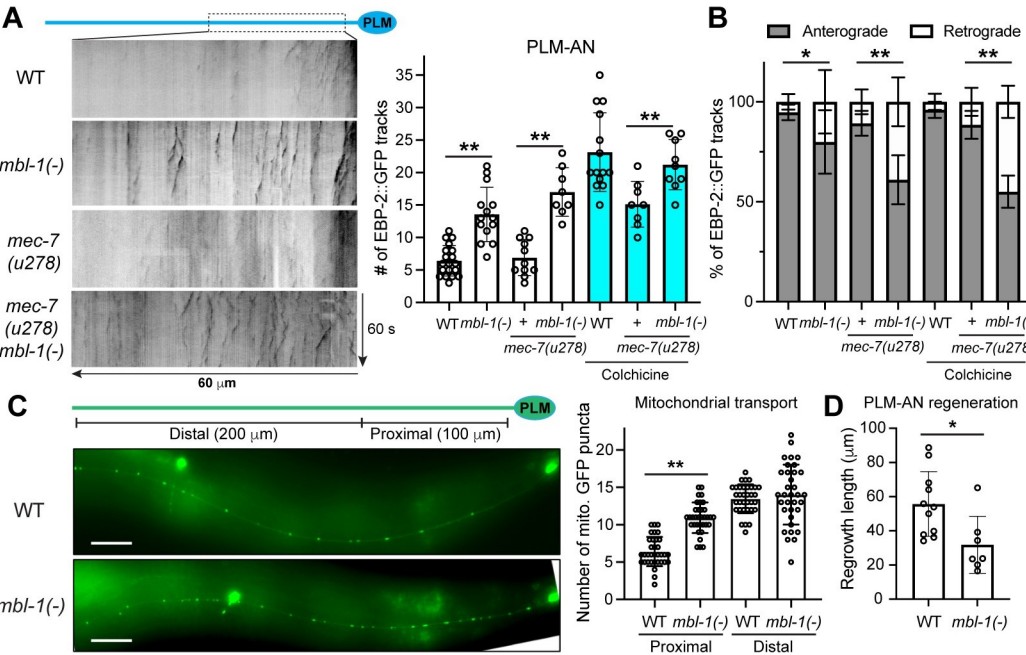

**Fig 2. MBL-1 regulates microtubule dynamics and mitochondrial transport.** (A) Representative kymographs of EBP-2:: GFP dynamics in the PLM-AN of various strains with quantification of the number of EBP-2 tracks. *mbl-1(tm1563)* was used as an *mbl-1(-)* allele. To increase MT dynamics, *mec-7(u278)* and *mec-7(278) mbl-1(-)* animals were subjected to a mild colchicine treatment and imaged after a one-hour recovery. (B) The percentage of EBP-2 comets that were anterograde and retrograde tracks, respectively, in various strains. Single and double asterisks indicate $p < 0.05$ and 0.01, respectively, in a post-ANOVA Tukey's HSD test. (C) Distribution of mitochondria in the PLM-AN visualized by the *jsIs609 [mec-7p*::*mitoGFP]* transgene and the quantification of the GFP puncta in the proximal and distal segments of the PLM-AN. Scale bar = 20 μm. (D) The regrowth length of PLM-AN in 24 hours after the laser axotomy in wild-type and *mbl-1(-)* animals. The asterisk indicates $p < 0.05$ in an unpaired *t*-test.

animals are highly stable and have very few EBP-2 tracks, we previously used a mild colchicine treatment (0.125 mM for 8 h) followed by a 1-h recovery to increase MT dynamics [22]. Under this sensitized condition, *mec-7(u278)* mutants had fewer EBP-2 tracks than the wild-type animals, but the loss of *mbl-1* increased the number of EBP-2 tracks in *mec-7(278)* mutants, restoring it to the wild-type level (Fig 2A).

Interestingly, the MT polarity was also altered in the *mbl-1(-)* mutants. In the wild-type and *mec-7(u278)* animals, MTs in the PLM-AN had largely uniform "plus-end-out" polarity [22], whereas 40–50% of the EBP-2 tracks observed in *mec-7(u278) mbl-1(-)* double mutants showed retrograde movement, indicating the "minus-end-out" MT orientation (Fig 2B). Our previous work found that mixed MT polarity in PLM-AN was often associated with increased MT dynamics and neurite growth defects [22], which is consistent with the phenotype in the *mbl-1(-)* mutants. The polarity of MTs was mildly disrupted in *mbl-1(-)* animals (Fig 2A and 2B). Overall, the above results indicate that MBL-1 controls neurite growth by promoting MT stability and regulating MT polarity.

In neurons, mitochondria are transported to the axons in a MT-dependent manner. So, we used mitochondrial distribution as an example to examine MT-mediated cargo transport and found that mitochondria appeared to be more accumulated in the proximal segment of the axons in *mbl-1(-)* mutants than the wild-type animals (Fig 2C). Transport to the distal segment also showed much higher variability in *mbl-1(-)* mutants compared to the more stereotypical localization of the mitochondria along the axons. These results indicate that the reduced MT stability affects cargo transport in *mbl-1(-)* mutants.

The altered MT stability in *mbl-1(-)* mutants also affected PLM axonal regeneration. After laser-induced axotomy, PLM-AN showed reduced regrowth length in *mbl-1(-)* mutants compared to the wild-type animals (Fig 2D), which is consistent with the idea that axonal regeneration requires persistent growth of stable MTs. Previous work found that the MT-destabilizing KLP-7/kinesin-13 inhibited axonal regeneration [23], and our results showed that the MT stabilizer MBL-1 promoted regeneration.

## MBL-1 has unique function in promoting neurite growth among three splicing regulators

Because Norris *et al.* found that *mbl-1* and *exc-7*, which codes for an ELAV-like RNA binding protein, acted redundantly to regulate lifespan [24], we tested whether *exc-7* also regulated TRN development. Mutations in *exc-7*, however, did not suppress the growth of ectopic ALM-PN in *mec-7(u278)* mutants, and *exc-7(-)* mutants had normal TRN morphology (S2 Fig). Moreover, *exc-7(-); mec-7(u278) mbl-1(-)* triple mutants were indistinguishable from *mec-7(u278) mbl-1(-)* double mutants for the suppression of ALM-PN growth and *exc-7(-); mbl-1(-)* double mutants did not exacerbate TRN defects compared to *mbl-1(-)* single mutants (S2B and S2C Fig).

Another splicing regulator *mec-8*, which codes for an ortholog of human RBPMS, was known to function in the TRNs to regulate the splicing of *mec-2* [25]. Moreover, *mec-8* and *mbl-1* also function redundantly to regulate the splicing of *sad-1* in ALM neurons, one of the TRN subtypes [26]. We found that mutations in *mec-8* did not suppress ALM-PN growth in *mec-7(u278)* mutants (S2B Fig). Since *mec-7* and *mec-8* are closely linked on chromosome X, to generate the *mec-7(u278) mec-8(-)* double mutants, we deleted *mec-8* through CRISPR/Cas9-mediated gene editing in the *mec-7(u278)* mutants (S2A Fig). Moreover, *mec-8(-)* single mutants did not show neurite growth defects in TRNs in the otherwise wild-type background (S2D Fig). These data indicate that although MBL-1 functions redundantly with other splicing regulators like EXC-7 and MEC-8 in some scenarios, the activity of MBL-1 in regulating neurite growth is not shared by the other two splicing factors.

## MBL-1 regulates the alternative splicing of neuronal genes and MAP genes

Next, we conducted transcriptomic profiling and exon junction analysis to identify MBL-1 target genes. Through RNA-sequencing, we compared the exon junction usage between the wild-type and *mbl-1(-)* mutants and between the *mec-7(u278)* single and the *mec-7(u278) mbl-1(-)* double mutants using the STAR (Spliced Transcripts Alignment to a Reference) software [27]. By finding the overlap of the junctions affected by mutations in *mbl-1* in both the wild-type and the *mec-7(u278)* backgrounds, we identified 57 downregulated and 30 upregulated junctions in 26 and 13 genes, respectively, in *mbl-1(-)* mutants (Fig 3A; S1 and S2 Tables). We then selected 16 genes that have either neuronal expression or functions related to MTs and confirmed the changes in their mRNA isoform levels in *mbl-1(-)* mutants using RT-PCR with primers spanning exon-exon junctions (Fig 3B). To test whether these MBL-1 target genes mediated the activity of MBL-1 in regulating neurite growth, we picked seven genes (*kin-4*, *lfi-1*, *mec-3*, *nid-1*, *nlp-38*, *ptl-1*, and *rbf-1*) that showed TRN expression either in the scRNA-seq dataset [28] or in promoter-reporter studies to conduct functional tests. *egl-8*, *frm-1*, and *unc-104* were not tested because they code for large proteins (>1000 amino acids) and the alternatively spliced isoforms produced small difference relative to the gene size. We either overexpressed the isoforms that were downregulated in *mbl-1(-)* mutants or inactivated the isoforms that were upregulated and examined whether the neurite growth pattern was affected.

For example, the loss of *mbl-1* downregulated the levels of the *a*, *b*, and *c* isoforms of the *ptl-1* gene by reducing the splicing between exon 1 and exon 2 (S3A and S3B Fig). *ptl-1* codes for a protein with Tau-like repeats and is the homolog of human MAP4; the *a*, *b*, and *c* isoforms contain an N-terminal proline and glutamic acid (PE)-rich domain, which does not exist in the *d* isoform. We overexpressed the *ptl-1 a* and *b* isoforms either separately or together in the *mec-7(u278) mbl-1(-)* double mutants but did not observe the growth of long ALM-PN. We also inactivated the *a*, *b*, and *c* isoforms by inserting a stop codon in the exon 2 of the endogenous *ptl-1* locus in the *mec-7(u278)* mutants but did not observe the suppression of ALM-PN growth (S3C Fig). Thus, we conclude that although MBL-1 regulates the splicing of *ptl-1*, the downregulation of the *ptl-1* isoforms may not be the cause of neurite growth defects in *mbl-1 (-)* mutants. Similar approach was used to test the functional consequence of abnormal alternative splicing of the other six genes (S3 Table), and we found that only the dysregulated splicing of the LIM homeodomain transcription factor *mec-3* had some effects on neurite development. The *a* isoform of *mec-3* was downregulated and the *d* isoform was upregulated in *mbl-1(-)* mutants compared to the wild-type animals (Fig 3C). The overall *mec-3* mRNA level was slightly increased in *mbl-1(-)* mutants probably due to the upregulation of *mec-3d*. This change in *mec-3* splicing pattern was validated using both semi-quantitative RT-PCR and quantitative real-time PCR (RT-qPCR) (Fig 3C and 3D). Overexpression of the *mec-3d* isoform in *mec-7 (u278)* mutants partially suppressed the ectopic growth of ALM-PN (Fig 3E), suggesting that at least part of the effects of MBL-1 on neurite growth may be mediated by the regulation of *mec-3* alternative splicing.

## Short isoforms of *mec-3* truncate the LIM domain and impair its function

Previous studies found that *mec-3* acts as a terminal selector for TRN fate specification and regulates the expression of a range of TRN differentiation genes [29,30]. Among the three annotated isoforms of *mec-3*, the *a* isoform is the longest, contains both exon 1 and exon 2, and codes for a protein with two LIM domains and one homeobox domain (Fig 4A). The shorter *d* isoform skipped exon 2, which led to the truncation of the first LIM domain with a deletion of the beginning 15 amino acids that included two conserved cysteine residues. The shortest *c* isoform included neither exon 1 nor exon 2, leading to the loss of the N-terminal 43

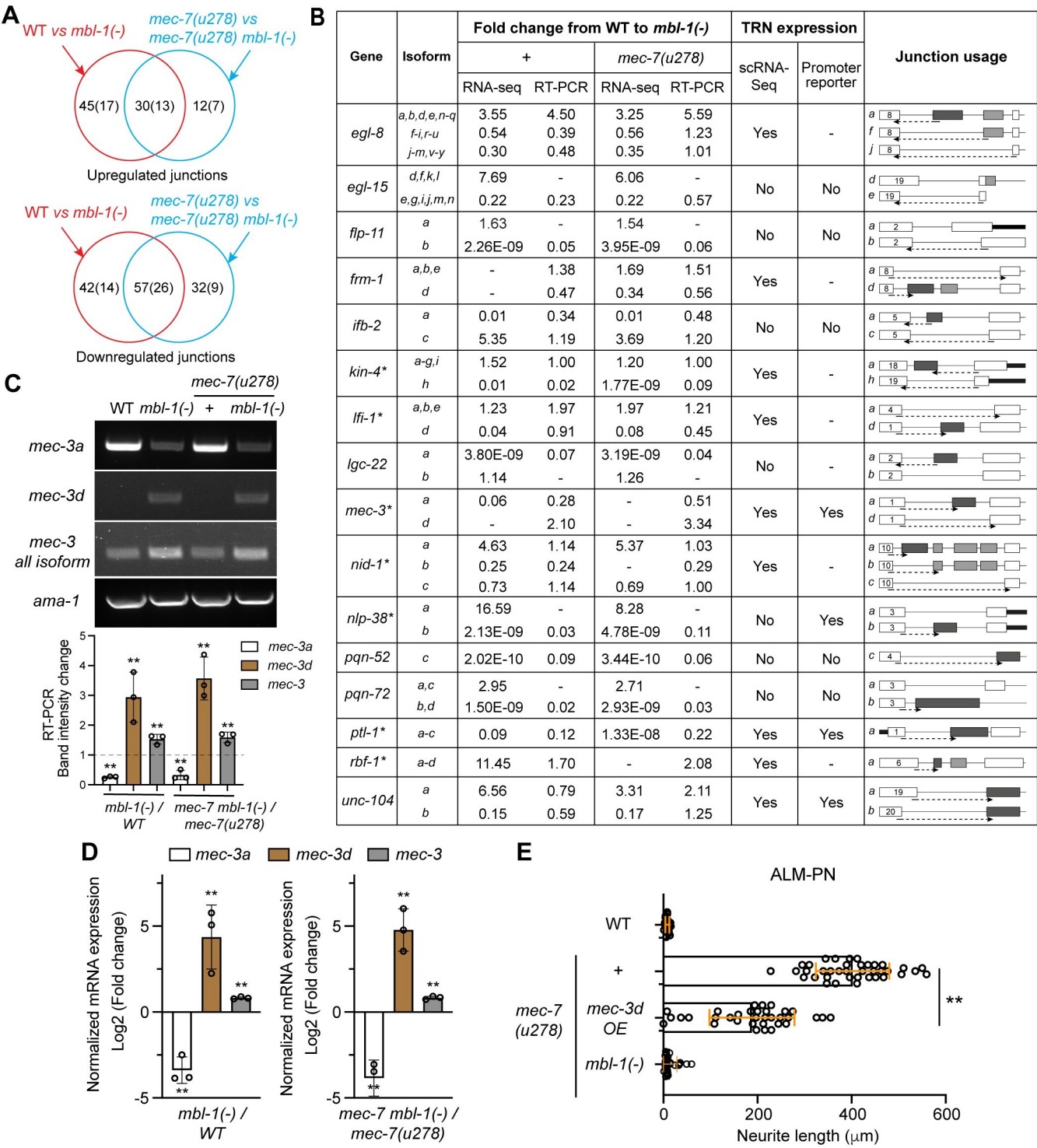

**Fig 3. Junction analysis identified mRNA splicing events controlled by MBL-1.** (A) The overlap between the significantly upregulated (or downregulated) exon-exon junctions in *mbl-1(-)* mutants compared to wild-type animals and the regulated junctions in *mec-7(u278) mbl-1(-)* double mutants compared to the *mec-7(u278)* single mutants. Details about the altered junction usage (adjusted *p* values < 0.05) can be found in S1 and S2 Tables. (B) Confirmation of altered splicing in *mbl-1(-)* mutants by reverse transcription-PCR (RT-PCR). Quantification of fold change in junction usage that is specific for certain isoforms was extracted from the STAR output of the RNA-seq results and then calculated based on the grey intensity of semi-quantitative RT-PCR experiments using primers that span isoform-specific exon-exon junctions (indicated by the arrow with dashed lines). To allow easy identification of the splicing junction, we added the exon number for the first exon shown for each isoform. Whether the gene has TRN expression was determined by the scRNA transcriptomic data [28] and the signals of promoter reporters. We constructed GFP reporters for *ifb-2*, *nlp-38*, *pqn-52*, and *pqn-72* in this study and used *ayIs2[egl-15p::GFP]* and

*ynIs40[flp-11p::GFP]* for *egl-15* and *flp-11*, respectively. TRN expression of *mec-3*, *ptl-1*, and *unc-104* were known from previous studies [56–58] and were annotated in Wormbase.org. The genes with asterisks were subjected for functional validation, and the results are shown in S3 Table. (C) RT-PCR results using isoform-specific forward primers for *mec-3a* and *mec-3d* and a common reverse primer or a pair of primers that bind to exon 3 and 6 that are shared by all *mec-3* isoforms. *ama-1*, encoding an RNA polymerase II subunit A, was used as the internal control. Quantification of the band intensity from three independent RT-PCR results (the ratio is shown). Double asterisks indicate $p < 0.01$ in a *t*-test comparing the wild-type and *mbl-1(-)* animals. (D) RT-qPCR results for the mRNA expression levels of *mec-3a*, *mec-3d*, and *mec-3* overall using isoform-specific or isoform-shared primers. -$\Delta\Delta C_T$ is shown as the fold change using *ama-1* as the internal control. (E) Quantification of ALM-PN length in *mec-7(u278)* animals with the overexpression of *mec-3d* cDNA from the *mec-3* promoter.

amino acids (Fig 4A). Since LIM domain is important for protein-protein interaction, we suspected that the partial deletion of one LIM domain in the short isoforms of MEC-3 may reduce its ability to regulate downstream genes. To test this hypothesis, we compared the activity of MEC-3 *a* and *d* isoforms in activating the TRN cell fate marker *mec-17p*::*TagRFP* by expressing the cDNA of these isoforms in *mec-3(-)* mutants. We found that MEC-3a was able to largely rescue the loss of the genomic *mec-3* and restore the TRN marker expression in all TRN subtypes, whereas MEC-3d showed very limited rescue of *mec-17* expression except for the PLM, which might have a lower threshold for TRN fate activation than the other subtypes (Fig 4B). Similar results were found for the expression of *mec-3p*::*GFP*, which relies on MEC-3 through autoregulation (Fig 4B). These findings confirmed that the LIM domain truncation caused by the skipping of exon 2 led to a lower level of activity in the short MEC-3 isoforms compared to the long isoform and that most functions of the *mec-3* gene were carried out by the *a* isoform.

To further support this idea, we deleted the exon 2 and its flanking sequences in the endogenous *mec-3* locus through gene editing to generate the *mec-3(syb8382)* mutants, which lack the *mec-3a* isoform; we also created the *mec-3a*-only allele *unk206* by deleting the first two introns and fusing exon 1, 2, and 3 together (Fig 4A). We confirmed that these alleles only produced the expected isoforms (S4A and S4B Fig). The TRN marker *mec-17p*::*TagRFP* was mostly expressed in the PLM neurons and was expressed at a much lower frequency in other TRN subtypes in the *mec-3(syb8382)* mutants, confirming that the *a* isoform is required for TRN fate specification (Fig 4C and 4D). Interestingly, *mec-3p*::*GFP* expression in PLM had low penetrance in *mec-3(syb8382)* animals, suggesting that the *d* isoform does not activate *mec-3* promoter through autoregulation (Fig 4C). In contrast, the *mec-3(unk206)* mutants showed the expression of TRN markers in 100% of the TRNs (Fig 4C and 4D), and the fluorescence level of the *mec-17p*::*TagRFP* is comparable to that in the wild-type animals (S4C Fig), suggesting that the *mec-3a* isoform alone is sufficient to drive TRN differentiation. For PLM neurons, the *mec-17p*::*TagRFP* and *mec-3p*::*GFP* reporters were expressed at a much higher level in *mec-3(unk206)* animals than in *mec-3(syb8382)* mutants when the expression were visible in both (S4C and S4D Fig), supporting that MEC-3a has a stronger ability to activate TRN genes than MEC-3d.

At the behavioral level, *mec-3(syb8382)* mutants were largely touch-insensitive at both the anterior and posterior, whereas *mec-3(unk206)* mutants were as sensitive as the wild-type animals (Fig 4E). The above results support that the MEC-3a isoform is required for the full activation of the TRN fate and their mechanosensory functions in all six subtypes.

## MBL-1 regulates the expression of TRN genes partly by controlling *mec-3* splicing

Given that *mec-3d* has reduced function compared to *mec-3a*, the switching from the *a* to *d* isoform in *mbl-1(-)* mutants may affect the activity of *mec-3* and cause defects in TRN differentiation. Indeed, from the RNA-seq data, we found that many known MEC-3-dependent TRN terminal differentiation genes, including the *mec-12*/α-tubulin, *mec-7*/β-tubulin, and *mec-17*/

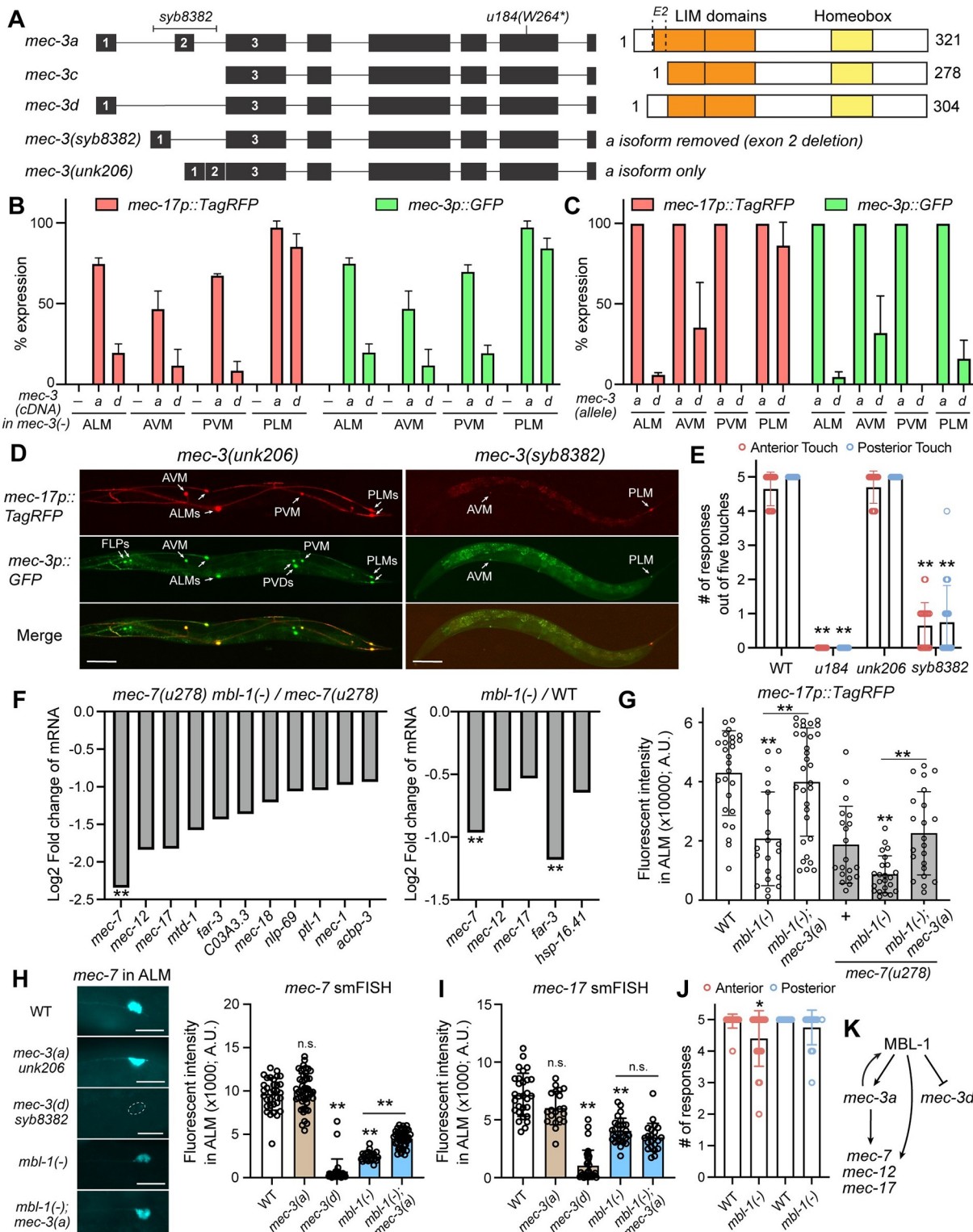

**Fig 4. MBL-1-controlled *mec-3* splicing regulates the TRN differentiation genes.** (A) Gene structure of *mec-3* and protein domain structures of MEC-3*a*, *c*, and *d* isoforms. Exon 2 (E2) encodes the 27–43 amino acids (a.a.) in the *a* isoform; the two LIM domains cover 29–83 a.a. and 89–146 a.a., respectively. *syb8382* allele deleted exon 2 and its flanking sequences (101-bp on the left and 125-bp on the right); *unk206* deleted intron 1 and intron 2 and fused exon 1, 2, and 3 together. (B) The percentage of TRN subtypes expressing the TRN fate markers *mec-17p*::*TagRFP* and *mec-3p*::*GFP* in *mec-3(u184)* animals expressing the *mec-3a* or *mec-3d* cDNA from the *mec-3* promoter. (C) The percentage of

TRN subtypes expressing the TRN fate markers in *mec-3(unk206)* and *mec-3(syb8382)* animals, labeled as *mec-3a* and *mec-3d* alleles. Over 45 animals were counted for each strain in (B) and (C). (D) Representative images of the TRN marker expression in the *mec-3* mutants. For *syb8382*, weak but visible signals were counted as showing expression. (E) The number of responses out of five touches at the anterior and posterior sides of the *mec-3* mutants. (F) Fold changes of gene expression values extracted from the RNA-seq results for some TRN-enriched genes, which were defined by previous studies [59]. All genes shown in the plot had $p < 0.05$ in pairwise comparison; the genes with double asterisks also had adjusted $p < 0.05$ after Benjamini-Hochberg corrections (details can be found in S4 and S5 Tables). (G) Fluorescent intensities of the promoter-reporter *mec-17p::RFP* in ALM neurons of various strains. (H) Representative images of smFISH staining against *mec-7* transcripts in the ALM neurons of the indicated strains and the quantification of the fluorescent signals in the cell bodies. Double asterisks indicate $p < 0.01$ and "n.s." indicates "not statistically significant" in comparison with the wild-type or between indicated pairs in a Tukey's HSD test. (I) Quantification of the smFISH signals against *mec-17* transcripts in the ALM cell bodies in various strains. (J) The number of responses out of five touches at the anterior and posterior sides of the *mbl-1(-)* mutants. The single asterisk indicates $p < 0.05$ in a *t*-test comparing wild-type and *mbl-1(-)* animals. (K) A proposed model for the role of MBL-1 in regulating the expression of TRN genes by promoting *mec-3a* splicing and the mRNA stability of TRN genes.

α-tubulin acetyltransferase, were downregulated in *mbl-1(-)* mutants (Figs 4F and S4E). This downregulation was further confirmed by the lower fluorescent intensity generated from the *mec-17p::TagRFP* transgene in the *mbl-1(-)* mutants (Fig 4G). Thus, we hypothesized that MBL-1 regulates TRN differentiation at least partly by promoting the splicing of the long and more active isoform of *mec-3*, which in turn activates the expression of downstream genes related to neuronal morphogenesis and functions. Supporting this hypothesis, the reduced expression of the *mec-17p::TagRFP* reporter in *mbl-1(-)* mutants was rescued by the non-spliceable *a*-isoform-only *mec-3(unk206)* allele (Fig 4G), suggesting that blocking MBL-1-mediated regulation of *mec-3* splicing could restore the activation of TRN genes.

To further understand how MBL-1 regulates TRN genes, we used single-molecule fluorescent *in situ* hybridization (smFISH) to detect endogenous *mec-7* and *mec-17* mRNAs and found that the abundance of their transcripts was significantly reduced in *mbl-1(-)* mutants in the TRNs (Fig 4H and 4I). To our surprise, this reduction was only partially rescued in *mbl-1(-); mec-3(unk206)* double mutants for *mec-7* and not rescued at all for *mec-17*, suggesting that correcting *mec-3* splicing defects were not sufficient to restore the mRNA levels of the TRN genes. Recently, a study by Puri *et al.* found that MBL-1 regulated the stability of *mec-7* and *mec-12* by directly binding to their mRNAs [21]; and Verbeeren *et al.* identified *mec-7*, *mec-12*, and *mec-17* among MBL-1-interacting RNAs in an RNA immunoprecipitation sequencing experiment [31]. We reasoned that MBL-1 may regulate the mRNA levels of TRN genes through two mechanisms: first, it acts as a splicing regulator to promote the expression of *mec-3a*, which activates the transcription of TRN genes; second, it acts as an RNA-binding protein to stabilize the mRNAs of certain TRN genes, particularly the ones related to microtubule functions (Fig 4K). Nevertheless, unlike *mec-3(syb8382)* animals, *mbl-1(-)* mutants were mostly touch sensitive (Fig 4J), suggesting that the low level of *mec-3a* and the reduced expression of TRN genes in *mbl-1(-)* mutants were still sufficient to drive most of the mechanosensory function of the TRNs. The slight reduction of anterior sensitivity and the normal posterior touch sensitivity in *mbl-1(-)* mutants (Fig 4J) appeared to be consistent with the stronger defects on the expression of TRN differentiation genes in ALM compared to that in PLM when *mec-3a* was replaced by *mec-3d* (Fig 4B and 4C). We noted that our results were different from the Puri *et al.* study, which found reduced touch sensitivity at both the anterior and posterior in *mbl-1(-)* mutants [21].

Interestingly, Thompson *et al.*, reported that *mbl-1* expression in the ALM neurons depended on *mec-3* [26]. We crossed the GFP reporters for different *mbl-1* isoforms (*mbl-1p1::GFP* and *mbl-1p2::GFP* in Fig 1F) with *mec-3* mutants and found that their TRN expression were lost in both *mec-3(-)* and *mec-3(syb8382)* mutants but not in the *mec-3(unk206)* mutants (S5A Fig). Similar results were found using the fosmid-based *mbl-1::EGFP* translational reporter (S5B Fig), indicating that the *mec-3a* was required for the expression of likely all

isoforms of *mbl-1* and that *mec-3d* alone was not sufficient to activate *mbl-1*. Since *mbl-1* promotes the splicing of *mec-3a* over *mec-3d*, our results suggest a positive feedback loop between *mbl-1* and *mec-3a* that may help initiate TRN differentiation (Fig 4K).

## MBL-1 regulates *mec-3* splicing through a canonical "YGCU(U/G)Y" motif

To understand how MBL-1 regulates *mec-3* splicing, we generated a dual-color splicing reporter by fusing the first three exons and two introns of *mec-3* with a DNA fragment that codes for GFP and RFP in two different reading frames (Fig 5A). When exon 2 is included, GFP will be out-of-frame and RFP will be in-frame, producing red and no green signal; when exon 2 is skipped, GFP will be in-frame and RFP will not be translated due to a stop codon between the GFP- and RFP-coding sequences. We first tested this splicing reporter in the wild-type animals and found that indeed the RFP signal was stronger than the GFP signal, indicating that the *a* isoform is the more dominant isoform compared to the *d* isoform. In *mbl-1(-)* mutants, the splicing reporter produced a much stronger GFP signal than RFP signal, suggesting a switch from including exon 2 to skipping it. The experiments were performed in the *smg-1(-)* background to prevent nonsense-mediated decay of mRNAs [32].

Previous studies found that the human MBNL1 recognizes the "YGCU(U/G)Y" motif in pre-mRNA [33–35]. We found three such motifs in and around exon 2 of *mec-3*; one (site 1) was within exon 2 and two (site 2 and 3) were in the intron downstream of exon 2 (Fig 5A). Deleting site 1 biased the splicing towards skipping exon 2, whereas deleting the other two sites had little effect. Deleting both site 1 and site 2 enhanced the preference towards skipping exon 2, while deleting all three sites created a splicing pattern identical to the *mbl-1(-)* mutants (Fig 5B and 5C). Thus, the three sites function redundantly to mediate the activity of MBL-1 in regulating *mec-3* splicing, with site 1 having the strongest effect followed by site 2 and then site 3. Moreover, the location of site 2 and 3 downstream of exon 2 whose inclusion was promoted by MBL-1 is consistent with the finding that binding of human MBNL1 to the intronic region downstream of exons generally enhanced inclusion of the exon [36].

The fact that *C. elegans* MBL-1 acts through the "YGCU(U/G)Y" motif recognized by human MBNL1 suggests that the RNA-binding specificity of the Muscleblind proteins is conserved across species and explains why the expression of human MBNL1 can rescue the loss of MBL-1 in the TRNs (Fig 1C).

## Correcting *mec-3* splicing defect is not sufficient to rescue neurite growth in *mbl-1(-)* mutants

Although *mbl-1(-)* mutants showed the defects in generating the MEC-3a isoform, transgenic overexpression of the *mec-3a* cDNA under its own promoter did not rescue either the PLM-AN growth defects in *mbl-1(-)* mutants or the loss of ALM-PN in *mec-7(u278) mbl-1(-)* double mutants (S6A and S6B Fig). We then considered the possibility that the *mec-3d* isoform upregulated in *mbl-1(-)* mutants may interfere with the effects of *mec-3a*. So, we overexpressed *mec-3a* in the *mec-3(-)* mutants to remove the potential interference from *mec-3d* and still did not observe any rescue on the neurite growth defects caused by the loss of *mbl-1*. Similarly, the non-spliceable *mec-3(unk206)* allele also failed to rescue the neurite extension defects when crossed with *mbl-1(-)* and *mec-7(u278) mbl-1(-)* mutants (S6A and S6B Fig), suggesting that correcting the *mec-3* splicing defect alone was not sufficient to restore neurite development in *mbl-1(-)* mutants. Supporting this result, the *mec-3a*-only *unk206* allele did not rescue the MT instability (measured by EBP-2 comets) in *mbl-1(-)* mutants either (S7C Fig).

Since overexpression of MEC-3d in *mec-7(u278)* mutants could cause ALM-PN shortening (Fig 3E) presumably by competing with MEC-3a for the same *cis*-regulatory sites in the

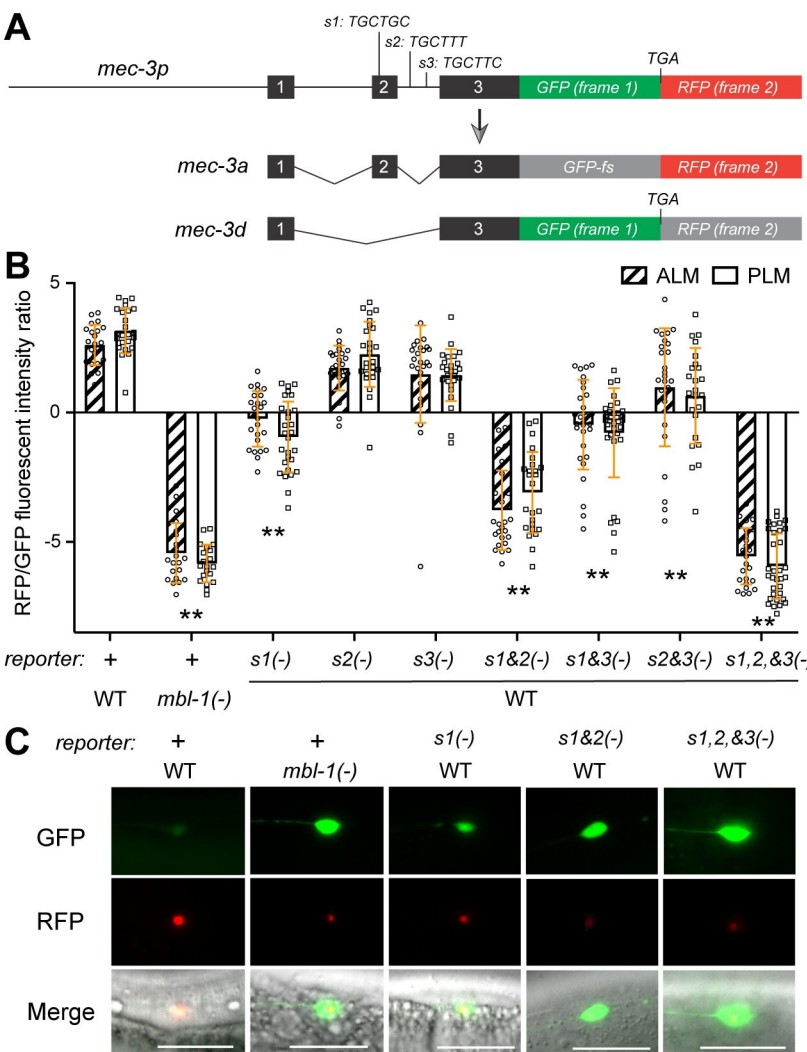

**Fig 5. MBL-1 regulates mec-3 splicing through "YGCU(U/G)Y" motifs.** (A) The structure of *mec-3* splicing reporter composed of *mec-3* genomic sequences from exon 1 to exon 3 fused to GFP and RFP sequences, which are in different coding frames. If exon 2 is included, GFP will be out of frame but still translatable and RFP will be in frame. If exon 2 is skipped, GFP will be in-frame and the translation will stop immediately after the GFP coding sequences. The putative MBL-1 binding motifs are shown as *s1*, *s2*, and *s3*. (B) Quantification of the ratio between RFP and GFP fluorescent intensities from the splicing reporter in either wild-type animals or *mbl-1(-)* mutants. "+" indicate that wild-type reporter was used. The "YGCU(U/G)Y" motifs were deleted either individually or in combination to create mutant reporters. Double asterisks indicate that *p* < 0.01 in a post-ANOVA Dunnett's test in comparison with the wild-type reporter in the wild-type animals. (C) Representative images of the splicing reporter or its variants in the ALM neurons in either the wild-type or *mbl-1(-)* animals. Scale bar = 20 μm.

downstream genes (the two isoforms share the same DNA-binding homeodomain), we reasoned that neurite extension in TRNs requires the activity of MEC-3a, but MEC-3a is not sufficient to induce neurite growth if the splicing of other MBL-1-regulated genes are still defective. So, we next attempted to test the MBL-1 target genes (identified in the RNA-seq data) in combinations by overexpressing the downregulated *mec-3a*, *ptl-1a/b*, *kin-4h*, *nlp-38b*, and *lfi-1d* isoforms together in *mec-7(u278) mbl-1(-)* mutants but still did not observe the recovery of long ALM-PN (S6D Fig). Given these results, we concluded that either the function of MBL-1 in promoting neurite growth is independent of its role as a splicing regulator or the function is mediated by coordinated action of many downstream effectors.

It should be noted that although Puri *et al.* reported the rescue of PLM-AN extension defects in *mbl-1(-)* mutants by overexpressing the tubulin genes *mec-7* or *mec-12* [21], we could not reproduce these results. In our hands, overexpression of either *mec-7(+)* or *mec-12 (+)* failed to rescue the PLM-AN underextension in *mbl-1(-)* mutants or the ALM-PN shortening in *mec-7(u278) mbl-1(-)* mutants (S6C and S6D Fig). We also overexpressed the *mec-7 (C303Y)* mutants coded by the *u278* allele in the *mec-7(u278) mbl-1(-)* mutants and did not observe any increase of the ALM-PN length (S6D Fig). In a final attempt, we overexpressed *mec-7(+)* in the *mbl-1(-); mec-3(unk206)* double mutants, in which the *mec-3* splicing defect was corrected and the normal activity of *mec-17* promoter (used to overexpress *mec-7*) was restored, and still did not rescue the PLM-AN growth defect (S6E Fig). Similarly, overexpression of either *mec-7(+)* or *mec-7(C303Y)* in *mec-7(u278) mbl-1(-); mec-3(unk206)* triple mutants did not restore ALM-PN growth (S6F Fig).

## MBL-1 regulates neurite growth by suppressing the DLK-1/p38 MAPK pathway

Since restoring *mec-3a* or overexpressing tubulin genes could not rescue the neurite growth defects in *mbl-1(-)* mutants, we searched for additional MBL-1 downstream genes through a forward genetic screen. We used the *mec-7(u278) mbl-1(-)* mutants as the starter strain and conducted a suppressor screen to find revertants that grew a long ectopic ALM-PN. We reasoned that if the loss of MBL-1 resulted in the activation of MT-destabilizers, loss-of-function mutations in these genes may suppress the effects of *mbl-1(-)* on neurite growth. We screened ~57,200 haploid genomes and isolated twelve suppressors and identified the phenotype-causing mutations in them through whole-genome resequencing. These twelve mutations were mapped to four genes, *dlk-1*, *pmk-3*, *uev-3*, and *cebp-1*, all of which belong to the MAP kinase pathway (Fig 6A–6C). *dlk-1* codes for a MAPKKK (homolog of MAP3K12), *pmk-3* codes for a p38 MAPK, *uev-3* codes for an E2 ubiquitin-conjugating enzyme variant that may activate *pmk-3*, and *cebp-1* codes for the transcription factor C/EBP homolog downstream of the MAPK signaling pathway. Mutations in any of these four genes inactivated the DLK-1/p38 MAPK pathway and rescued the growth of ALM-PN in *mec-7(u278) mbl-1(-)* double mutants (Fig 6B and 6D). Nevertheless, the rescue was partial since the ALM-PN length in the suppressor mutants were shorter than that in the *mec-7(u278)* animals. Mutations in *dlk-1* also partially suppressed the MT instability in *mbl-1(-)* mutants, although the loss of either *dlk-1* or *uev-3* did not significantly rescue the PLM-AN growth defect in *mbl-1(-)* animals (S7A-S7C Fig). We reason that inactivating the DLK-1 pathway may not be sufficient to suppress all defects in *mbl-1(-)* mutants, and its effects were more obvious in the *mec-7(u278)* sensitized background. Overall, our results suggested that the loss of *mbl-1* may lead to abnormal upregulation of MAPK signaling, which may in turn result in neurite growth defects due to reduced MT stability. This result is consistent with previous findings that DLK-1 and the MAPK pathway promotes MT dynamics during axonal regeneration [23]. Intriguingly, the same DLK-1/ p38 MAPK pathway was also found downstream of RPM-1 in regulating axonal termination in PLM-AN. Mutations in *rpm-1*, which codes for an atypical RING E3 ubiquitin ligase, resulted in a PLM-AN overextension phenotype, which could be rescued by inactivating any of the six genes (*dlk-1*, *mkk-4*, *pmk-3*, *mak-2*, *uev-3*, and *cebp-1*) in the pathway [37–39]. Since the *rpm-1(-)* phenotype is the opposite of the PLM-AN underextension phenotype observed in the *mbl-1(-)* mutants, we constructed a *rpm-1(-); mbl-1(-)* double mutants and found that the double mutants showed an underextension phenotype similar to that in the *mbl-1(-)* single mutants (Fig 6E). Thus, although the same DLK-1/p38 MAPK pathway operated downstream of both RPM-1 and MBL-1, the effects of MBL-1 appeared to be more dominant. This

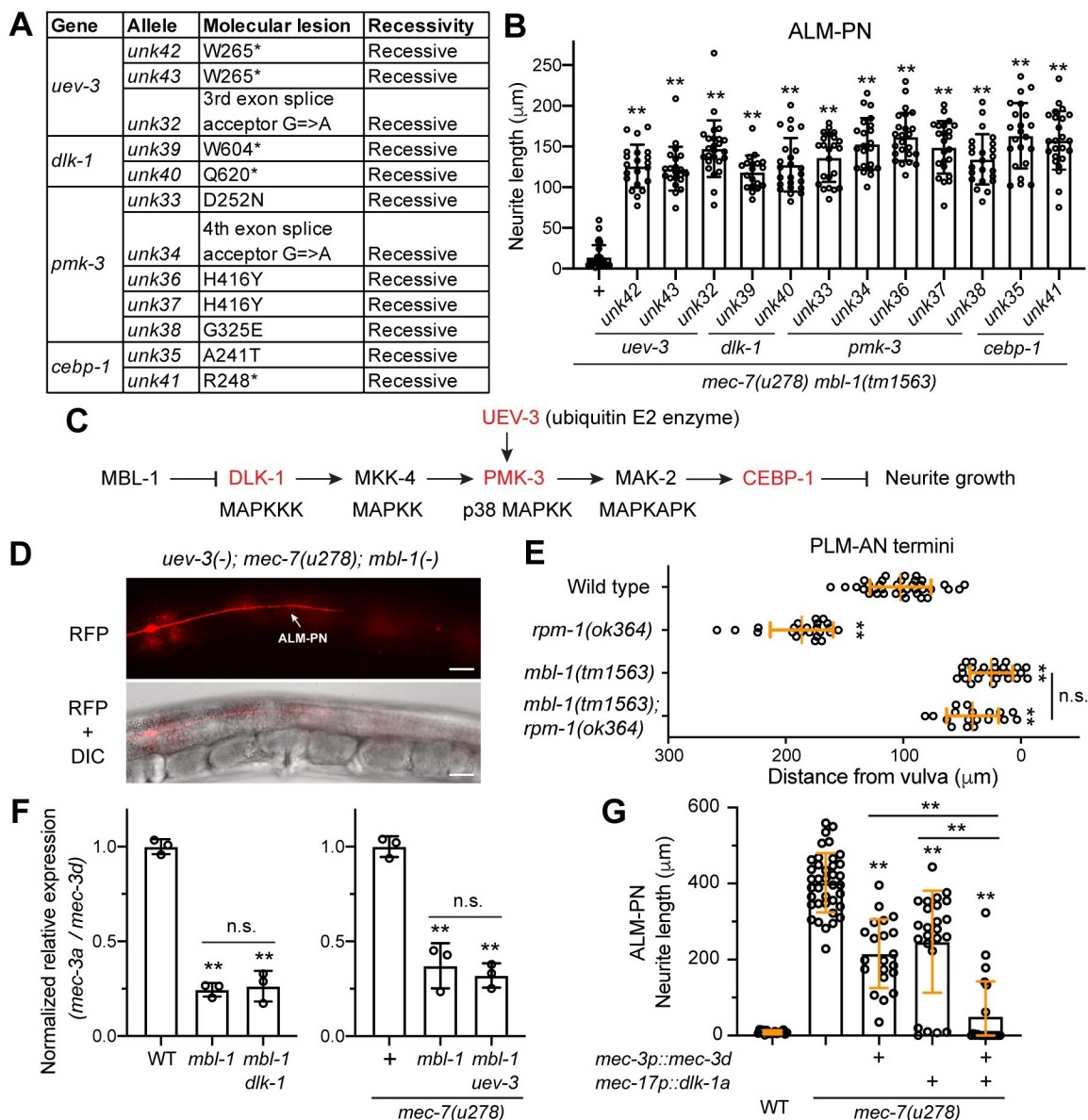

**Fig 6. MBL-1 promotes neurite growth by inhibiting the DLK-1 pathway.** (A) A summary of the alleles isolated from the suppressor screen using TU6020 *mec-7(u278) mbl-1(tm1563); uIs115[mec-17p::TagRFP]* as the starter strain and searching for mutants with long ALM-PN. (B) Quantification of ALM-PN length in the revertants isolated from the screen. Double asterisks indicate that $p < 0.01$ in a post-ANOVA Dunnett's test in comparison with the starter strain. (C) The genetic interaction between MBL-1 and the DLK-1 pathway in regulating neurite growth. The DLK-1 pathway was established by previous studies [37–39]; mutations in the genes in red were identified in our suppressor screen. (D) A representative image of the restored long ALM-PN in the *uev-3(unk61); mec-7(u278) mbl-1 (tm1563); uIs115* animals; *unk61* is a *uev-3* deletion allele we created through CRISPR/Cas9-mediated gene editing (see S7 Fig). Scale bar = 20 μm. (E) Quantification of PLM-AN length by the distance between the PLM-AN terminus and the vulva in various mutants. (F) Results of reverse transcription quantitative PCR (RT-qPCR) using isoform-specific primers for the comparison of the ratio of *mec-3a* to *mec-3d* mRNA levels in the various mutants. (G) ALM-PN length in *mec-7(u278)* mutants that overexpressed *mec-3d* or *dlk-1a* individually or in combination. In (E-G), double asterisks indicate that $p < 0.01$ in a Tukey's HSD test, and "n.s." indicates no statistical significance in the tests for the pair.

dominance might be because RPM-1 acts locally at the axonal terminus, whereas MBL-1 functions more globally in the entire neuron by regulating mRNA metabolism.

From the RNA-seq data and independent RT-qPCR tests, we did not find any changes in the splicing pattern or the expression levels of the six genes in the DLK-1/p38 pathway in *mbl-1(-)*

mutants, suggesting that they may not be the direct targets of MBL-1. Instead, MBL-1 may indirectly control the activity of the pathway. Nevertheless, we could not rule out the possibility that the MAPK pathway acts in parallel to antagonize the activity of MBL-1 in promoting neurite growth.

Next, we asked whether mutations in the DLK-1/p38 MAPK/CEBP-1 pathway suppressed the *mbl-1(-)* phenotype by correcting the *mec-3* splicing defects in *mbl-1(-)* mutants. Through RT-qPCR, we found that the relative expression of the *mec-3a* isoform to *mec-3d* isoform is similar between the *mbl-1(-)* single mutants and the *mbl-1(-); dlk-1(-)* double mutants and between the *mec-7(u278) mbl-1(-)* double and *mec-7(u278) mbl-1(-); uev-3(-)* triple mutants (Fig 6F), suggesting that the DLK-1 signaling did not affect *mec-3* splicing. Moreover, we found that overexpression of *dlk-1a* (the long isoform) led to the suppression of ALM-PN growth in *mec-7(u278)* mutants, supporting that DLK-1 functions to destabilize MTs. Importantly, overexpression of both *dlk-1a* and *mec-3d* led to stronger suppression of ALM-PN than either alone in *mec-7(u278)* mutants (Fig 6G), which is consistent with the idea that the DLK-1/p38 MAPK pathway and the regulation of *mec-3* splicing work in parallel to regulate TRN neurite morphogenesis. Nevertheless, *mec-3a*-only *unk206* allele did not enhance the effects of *dlk-1(-)* in promoting ALM-PN growth in the *mec-7(u278) mbl-1(-)* background (S7D Fig), suggesting that there are still other molecular defects that need to be corrected in the *mbl-1(-)* mutants to fully restore the normal neurite growth capacity.

### MBL-1-controlled *mec-3* splicing regulates dendrite arborization in PVD neurons

In addition to regulating TRN differentiation, MEC-3 is also known to promote dendritic arborization in the multi-dendritic sensory neuron PVD [40]. Because MBL-1 regulates the splicing of *mec-3*, we hypothesized that MBL-1 may also contribute to dendrite development in PVD by controlling the expression of *mec-3* isoforms. First, we found that the numbers of secondary, tertiary, and quaternary dendrites were all reduced in both *mec-3(-)* and *mec-3(syb8382)* mutants, in which the exon 2 of *mec-3* was deleted (Figs 7 and S8). These results suggest that *mec-3a* isoform is required and the *d* isoform is not sufficient for dendrite morphogenesis. Second, *mbl-1(-)* mutants had significantly fewer quaternary dendrites compared to the wild-type animals, although their numbers of secondary and tertiary dendrites were similar (Fig 7), indicating that MBL-1 promotes the growth of terminal dendritic arbors in PVD neurons. Moreover, the non-spliceable *a* isoform-only *mec-3(unk206)* allele significantly rescued the loss of quaternary dendrites in the *mbl-1(-)* mutants, indicating that MBL-1 regulates dendritic development by promoting the splicing of *mec-3a* isoform. Thus, our work found that MBL-1 promotes the growth of both axons and dendrites in *C. elegans*. Unlike its functions in TRN axonal extension, which probably rely on multiple downstream effectors, the activity of MBL-1 in PVD dendritic development is mostly mediated by the control of alternative splicing of *mec-3*.

While we are preparing this manuscript, Xie *et al.* also reported that MBL-1 regulates the dendritic complexity of PVD neurons and found that MBL-1 regulates *mec-3* splicing through a candidate approach [41]. Our results are consistent with their findings, although we arrived at *mec-3* through an unbiased transcriptomic approach and exon-exon junction analysis (Fig 3).

## Discussion

### Muscleblind regulates neurite morphogenesis and synaptic formation across species

Our study and two other recent studies [21,41] in *C. elegans* and previous work in *Drosophila*, mice, and human cells revealed an evolutionarily conserved function of Muscleblind proteins

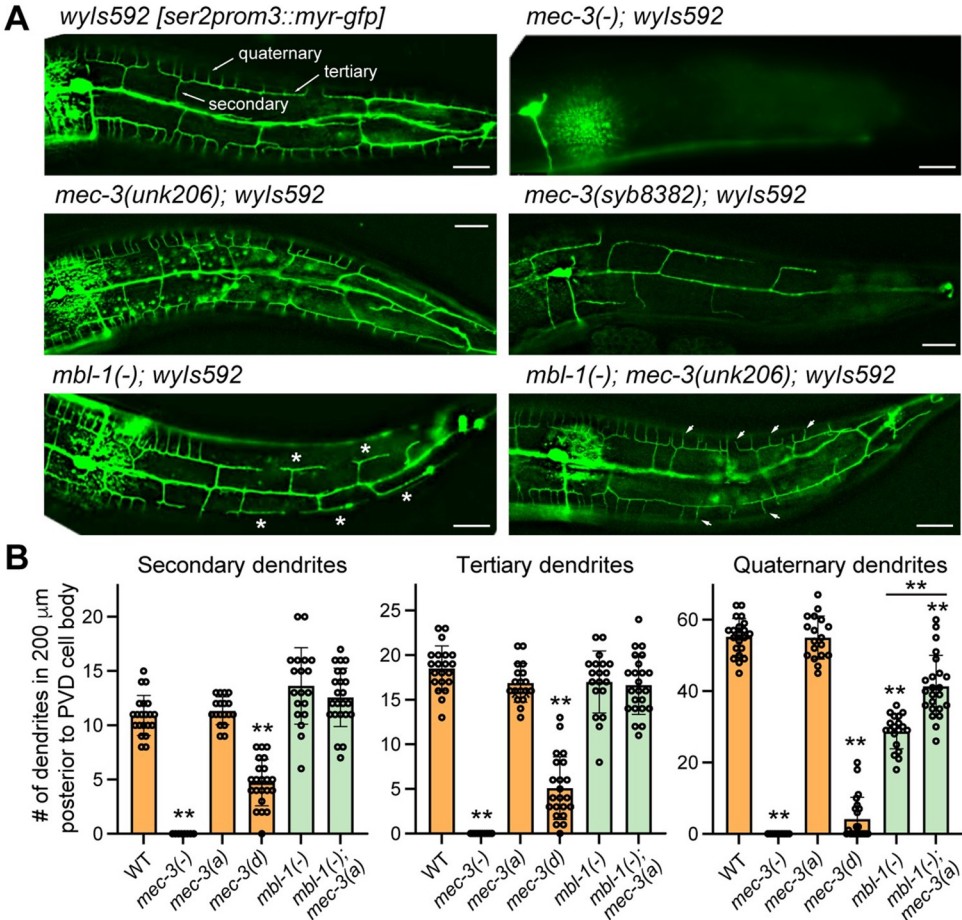

**Fig 7. MBL-1 regulates dendritic arborization by promoting *mec-3a* splicing.** (A) Representative images of PVD morphologies posterior to the cell body in various strains. Arrows in the wild-type image point to the secondary, tertiary, and quaternary level dendrites. Asterisks in the *mbl-1(-)* mutant image indicate the absence of quaternary dendrites on these tertiary dendrites. Arrow heads in the *mbl-1(-); mec-3(unk206)* mutants indicate the restoration of the quaternary dendrites. (B) Quantification of the number of secondary, tertiary, and quaternary dendrites in a region posterior to the PVD cell body but ≤ 200 μm away from the cell body. Images and quantifications of the dendrites anterior to the PVD cell body are shown in S8 Fig.

in regulating neurite morphogenesis and synaptic formation. In mice, both brain-specific knockout of MBNL1 and MBNL2 and brain-specific expression of expanded CUG RNA in the myotonic dystrophy model resulted in the reduction of dendritic complexity and alteration of postsynaptic morphology [17,42]. MBNL1 also promotes neurite outgrowth in cultured hippocampal neurons [15]. In *Drosophila* lamina neurons, the loss of Mbl led to reductions in axon arbor size and dendritic array width [18]. In *C. elegans*, MBL-1 promotes axonal growth in the mechanosensory TRNs and dendritic arborization in the nociceptor PVD neurons (this study and [21,41]) and regulates synaptic formation in the DA9 motor neuron [19]. Thus, Muscleblind appears to exert similar functions in multiple neuron types across species. Importantly, the human MBNL1 was able to rescue the loss of *C. elegans* MBL-1 (this study) and the embryonic lethality of Drosophila *Mbl* mutants [40], supporting functional conservation throughout evolution.

The effects of Muscleblind on neurite formation and synaptogenesis are consistent with the functions of its target genes. Previous CLIP-seq analysis found that MBNL1-bound brain

mRNAs showed significant enrichment in gene ontology categories, such as axon, neuronal projection, synapse, and synaptosome [43]. More recently, Weyn-Vanhentenryck *et al.* analyzed the temporal regulation of alternative splicing in the genome throughout neuronal development and found that MBNL1/2 (together with Rbfox1-3 and Nova1/2) mostly promoted the mature neuron-specific splicing pattern of developmentally regulated exons, whereas PTBP1/2 suppressed the mature splicing pattern [44]. Interestingly, among the four splicing regulators, the targets of MBNL1/2 switched isoforms at the latest developmental stage (P7 or older), supporting the idea that Muscleblind proteins mostly regulate late events of neuronal maturation, such as neurite morphogenesis and synaptic formation.

How does Muscleblind regulate neurite outgrowth? Our work established a connection between Muscleblind and the regulation of MT stability and polarity. Given that MTs not only provide structural support for neurite extension but also respond to guidance cues during axonal pathfinding, the modulation of MT dynamics enables Muscleblind to promote neurite growth. In fact, our transcriptomic analysis identified several genes encoding MT-associated proteins (MAPs) among the MBL-1 targets. Further investigations are needed to understand whether and how their alternative splicing affects MT properties.

## MBL-1 contributes to terminal differentiation by regulating the splicing of terminal selectors

The finding that MBL-1 regulates the splicing of *mec-3* was unexpected because *mec-3* is one of the terminal selectors that controls TRN fate specification and morphogenesis by activating its terminal differentiation genes, including the TRN-specific tubulins (*mec-12*/α-tubulin and *mec-7*/β-tubulin) and MT-related genes (e.g., *mec-17*/α-tubulin acetylase and *ptl-1*/Tau). MBL-1 is required to maintain a sufficient level of MEC-3 activity by promoting the splicing of the *mec-3* long isoform over the shorter isoform, which has a truncated LIM domain and is less active. In *mbl-1(-)* mutants, the switch from *mec-3* long to short isoforms led to the reduced transcription of TRN genes and impaired neuronal morphology and functions. Similarly, in the PVD neurons, MEC-3 promotes dendritic branching by regulating *hpo-30*/Claudin-like membrane protein [45], and MBL-1 regulates *mec-3a* splicing to ensure its activity during dendritic arborization.

Interestingly, we and others found that the expression of *mbl-1* in the TRNs was also dependent on MEC-3 [26]. In *mec-3(-)* mutants, the anterior TRN subtype ALM neurons adopted the fate of its sister cells (BDU neurons), and *mbl-1* expression was lost in ALM [26, 46]. These results suggest a positive feedback loop during TRN terminal differentiation—terminal selectors activate RNA splicing regulators, which in turn promote the productive splicing of the terminal selector. Moreover, in TRNs, MBL-1 and MEC-8/RBPMS act redundantly to regulate the inclusion of exon 15 of *sad-1* (a gene involved in synapses) which does not occur in the BDU neurons, suggesting that MBL-1 also contributes to neuronal fate specification by generating neuron type-specific splicing patterns. Similarly, Muscleblind regulates cell type-specific splicing of *Dscam2* in the *Drosophila* photoreceptors and lamina neurons [18]. Overall, the above findings highlighted the importance of RNA splicing factors in regulating neuronal fate specification.

## Distant Muscleblind homologs recognize the same RNA motifs

Functional conservation of Muscleblind proteins across species may stem from the fact that all Muscleblind homologs recognize similar RNA motifs that contain the "YGCY" core sequence. Early studies of human MBNL1 target genes, *Troponin T* (TNNT2 and TNNT3) and SERCA1, defined a "YGCU(U/G)Y" motif recognized by Muscleblind [33–35]. Subsequent SELEX

analysis found that MBNL1 preferentially bound to pyrimidine-rich RNAs containing "YGCY" motifs [36], while Drosophila Mbl recognizes a similar "YGCY" motif with a preference for at least one Y being U [47]. CLIP-seq and RNA Bind-n-Seq results identified more specific and sub-optimal motifs for MBNL1, including "GCUU" and "UGCU" (or "YGCU") [43,48]. *C. elegans* MBL-1 recognized both a "YGCU(U/G)Y" motif in *mec-3* (this study) and a "GCUU" motif in *sad-1* [26]. Thus, Muscleblind in humans, Drosophila, and *C. elegans* shared the same sequence preferences, which explains why human MBNL1 can functionally replace *Drosophila* Mbl and *C. elegans* MBL-1 in genetic experiments. Oddo *et al.* [49] further showed that distant Muscleblind homologs in *Ciona intestinalis* (tunicate) and *Trichoplax adhaerens* (placozoa) could also regulate alternative splicing and likely recognize similar RNA sequences as human MBNL1, suggesting that the splicing activity of Muscleblind proteins is deeply conserved.

Muscleblind proteins also showed context-dependent splicing regulation. Binding of MBNL1 to the upstream or within the 3' splice site generally represses splicing (leading to the exclusion of the exon), while binding to the downstream of the exon activates splicing (leading to the inclusion of the exon) [33,36]. This rule also applies to *C. elegans*, as we found the MBL-1 motifs located downstream of the *mec-3* exon 2 promoted its inclusion in the long isoform. Thus, our results are consistent with the Oddo study [49], which assayed five distant Muscleblind homologs in mouse embryonic fibroblasts for their splicing activity and found similar context dependency.

### Functional effectors of Muscleblind in neurons

One of the challenges in studying RNA splicing factors is the difficulty of identifying functional effectors downstream of their activities. Although RNA-seq and CLIP-seq analyses identified numerous Muscleblind target genes in a range of tissues, very few of them were functionally validated as the main effector of Muscleblind. In this study, we identified over 50 MBL-1-regulated genes but manipulating the isoform levels did not produce any detectable effects on neurite growth for most genes. For example, neither overexpression of the downregulated isoforms (e.g., overexpression of *ptl-1a-c* or *mec-3a*, etc.) nor correcting the splicing error (e.g., non-spliceable *mec-3a*) rescued neurite growth defects in the *mbl-1(-)* mutants. Even the inactivation of the DLK-1/p38 MAPK pathway could only partially rescue the loss of *mbl-1*. We reason that Muscleblind may regulate cellular behaviors and developmental processes through multiple pathways, and restoration of the splicing pattern of only one or a few genes may not be able to rescue the loss of Muscleblind. Future analyses of the downstream genes may have to simultaneously manipulate the isoform ratios of many genes to produce combinatorial effects that are similar to the overall effects of Muscleblind.

Alternatively, Muscleblind may also have functions that are independent of its activities in regulating alternative splicing. For example, human MBNL proteins are known to regulate RNA localization and stability by binding to their 3'UTR [43,50]. In *C. elegans*, Puri *et al.*, [21] and Verbeeren *et al.*, [31] found that MBL-1 can directly bind to many mRNA transcripts, including the TRN-specific *mec-7*, *mec-12*, and *mec-17* mRNAs, to potentially stabilize these mRNAs. Thus, the splicing-independent activities may also mediate the function of Muscleblind in neuronal development.

### Materials and methods

#### Strains, DNA constructs, and transgenes

*C. elegans* wild-type (N2) and mutant strains were maintained as previously described [51]. Most of the experiments were performed at 20˚C unless otherwise indicated. The *mbl-1*

*(tm1563)* and *mec-3(u184)* were used as *mbl-1(-)* and *mec-3(-)* null alleles, respectively, throughout this study. Lists of strains, DNA constructs, and primers used in this study can be found in S6 Table.

For the *mbl-1* promoter-reporter constructs, *mbl-1p1::GFP* (CGZ#112) and *mbl-1p2::GFP* (CGZ#121) were made by cloning a 4.9 kb and a 4.6 kb promoter sequence upstream of the start codons of *mbl-1 a* and *f* isoforms respectively into the pPD95.75 backbone. Similarly, a 4,525-bp *ifb-2* promoter, a 3,856-bp *nlp-38* promoter, a 1,718-bp *pqn-52* promoter, and a 5,000-bp *pqn-72* promoter were cloned and inserted to the upstream of GFP to create other reporters. For TRN-specific expression constructs, the genomic DNA sequence of *mbl-1* (encoding the *a*, *b*, and *c* isoforms), *mec-7*, *mec-7(C303Y)*, and *mec-12*, and the cDNA sequences of *kin-4h*, *lfi-1a* and *d*, *nid-1a*, *b*, and *c*, *nlp-38b*, *ptl-1a* and *b*, and *rbf-1a* isoforms were cloned into a vector containing a 1.9 kb *mec-17* promoter using the Gibson Assembly method with the ClonExpress kit from Vazyme Biotech (Nanjing, China). Similarly, the cDNA of human MBNL1 (transcript variant 21; NM_001376829.1 from NCBI) was cloned and inserted to the downstream of the *mec-17* promoter. These DNA constructs were injected into the gonads of worms to make transformants that carry extrachromosomal arrays.

Transgenes *uIs115[mec-17p::TagRFP] IV*, *uIs134[mec17p::TagRFP] V*, *uIs31[mec-17p::GFP] III*, and *uIs105[mec-3p::GFP]* were used to visualize TRN morphology. Transgene *wyIs592[ser-2prom3::myrGFP, odr-1::dsRed] III* was used to visualize PVD morphology. Transgene *jsIs609 [mec-7p::mtGFP + lin-15(+)] X* was used to visualize mitochondrial localization in the TRNs. The transgene *juIs338[mec-4p::ebp-2::GPF + ttx-3p::RFP]* was used to monitor MT dynamics.

## CRISPR/Cas9-mediated gene editing

To create deletion alleles for *uev-3* and *mec-8*, we used CRISPR/Cas9-mediated gene editing to make cuts at two separate sites of the endogenous locus. Specifically, a pair of single guide RNAs (sgRNAs) were synthesized using the EnGen sgRNA Synthesis Kit (E3322V) from NEB; a total of 1μg of the sgRNA pair along with 20 pmol recombinant Cas9 (EnGen S. pyogenes Cas9 NLS from NEB, M0646T) were injected into the *C. elegans*. pCFJ104 (*myo-3p::mCherry*) was used as a co-injection marker, and the transformants with red muscles were genotyped for successful deletion of the targeted gene.

To make precise editing of the *ptl-1* and *mec-3* loci, we used a previously published gene editing protocol that utilizes 0.1 μg/μl single-stranded DNA oligos as the repair donor for homologous recombination [52]. Synonymous mutations were added to the repair template to prevent digestion by Cas9. All CRISPR alleles were created in our lab except for *mec-3 (syb8382)*, which was generated by SunyBiotech (Fuzhou, China). CRISPR target sequences and repair donor sequences used to generate the alleles can be found in S6 Table.

## *mbl-1(-)* suppressor screen

For the *mbl-1(-)* suppressor screen, we used TU6020 [*mec-7(u278) X; mbl-1(tm1563) X; uIs115 (mec-17p::RFP) IV*] as the starter strain in which the long ALM-PN phenotype was suppressed. Ethyl methanesulfonate (EMS) was used as the mutagen [51]. After screening through 57,200 haploid genomes, we isolated twelve mutants that showed rescue of the ALM-PN growth. To map the phenotype-causing mutation, we outcrossed the mutants against the starter strain for at least six times to remove background mutations and identified candidate mutations using whole-genome resequencing. We then confirmed the phenotype-causing mutation by complementation tests against known loss-of-function alleles of the gene.

## Microtubule dynamics analysis and axonal regeneration

To study the effects of MT dynamics, we tracked EBP-2::GFP comets in the TRNs using the *juIs338[mec-4p::ebp-2::GPF + ttx-3p::RFP]* transgene based on a previous protocol [22]. Day-one adults were mounted with 100 nm polystyrene beads (Polysciences #00876–15; 5-fold dilution) on an agarose pad, and their PLMs were imaged for 1 min with 150-ms exposure time and 350-ms time intervals. The number of EBP-2::GFP tracks was recorded within the region 60 μm anterior to the PLM cell body. For the colchicine-treated studies, L4 animals were treated for 8 hours on a seeded NGM plate containing 0.125 mM colchicine and recovered on a regular NGM plate for 1 hour before being mounted for imaging.

For axonal regeneration, late-L4 animals were mounted on 10% agarose pads using polystyrene beads, and laser axotomy was performed using a Pulsed Laser Unit attached to the Infinity Scanner on a Leica DMi8 using 63x water lenses. To reduce the probability of reconnection, we made two cuts in PLM-AN at 40 and 80 μm anterior to the PLM cell body, respectively. Animals were then rescued and placed on NGM plates for recovery and were imaged 24 hours after the axotomy. The regrowth length was measured from the point of the first cut (40 μm anterior to the PLM cell body) to the terminals of the regrowing axon.

## Transcriptomic analysis and junction analysis

Total RNA was extracted from L4 animals using TRIzol reagent (Thermo Fisher). Two biological replicates were prepared for each strain. RNA samples were sent to BGI (Beijing Genome Institutes, Shenzhen, China) for standard library construction and pair-end sequencing. Around 20 million reads were obtained for each sample, and the reads were aligned to the *C. elegans* genome (WS235) using STAR 2.7 [53]. The splice junction output was used to identify exon-exon junctions that have statistically significantly different usages between the wild-type and *mbl-1(-)* animals using the Benjamini-Hochberg correction of the *p* values. DESeq2 was used to identify genes differently expressed between the wild-type and *mbl-1(-)* animals. The RNA-seq datasets are available at NCBI Gene Expression Omnibus (GEO) with the accession number GSE274472.

To verify the accuracy of the transcriptomic and junction analysis, cDNA libraries were reversed transcribed from the total RNA using SuperScript II Reverse Transcriptase with oligo (dT)s (Thermo Fisher). Potential candidates were selected for semi-quantitative RT-PCR using isoform-specific primers for independent validation (Fig 3). In some cases, quantitative real-time PCR (RT-qPCR) was performed using the cDNA libraries as the template and a TB Green Premix Ex Taq kit (Takara) in a CFX96 real-time PCR machine (BioRad). All initial values obtained were normalized to the internal control of *ama-1*, and the fold change was calculated by comparison with the normalized wild-type control. Each data point presented represents the average of three biologically independent replicates. The isoform-specific and qPCR primers used are shown in S6 Table.

## Single-molecule RNA fluorescent *in situ* hybridization (smFISH)

To quantitatively measure the abundance of *mec-7* and *mec-17* mRNA in the TRNs, we designed and synthesized Stellaris FISH Probes against the *mec-7* and *mec-17* transcripts using the Stellaris RNA FISH Probe Designer (www.biosearchtech.com/stellarisdesigner) from Biosearch Technologies (Petaluma, CA). Following the manufacturer's instructions and a previous protocol [54], L1/L2 animals of various strains were fixed, stained with the Stellaris RNA FISH Probe set labeled with Cy5 in the hybridization buffer, and then washed with the wash buffer purchased from Biosearch Technologies. Hybridized animals were then mounted using VECTASHIELD Antifade Mounting Medium (Vector Laboratories #H-1000) and imaged

using Leica DMi8 with Cy5 filter sets. Single mRNA transcripts cannot be easily resolved due to the high expression levels of *mec-7* and *mec-17*. So, the fluorescent intensity of the FISH staining in the ALM and PLM cell bodies were quantified.

## Gentle touch sensitivity

To test the gentle touch response, animals were stroked with an eyebrow hair as previously described [55]. A total of 20 day-one adults were tested for each strain with five touches each to the anterior and posterior regions of the animals. 5-second intervals were placed between each touch. Animals subjected to anterior touches will not be used for posterior touches, and *vice versa*. Positive touch response was recorded when the animal moved backward or forward within one second of the anterior or posterior touch, respectively. All touch assays were conducted in a blinded condition, ensuring that the results were free from bias.

## Dual-color splicing reporters

To construct the bi-fluorescent reporter plasmid, a DNA sequence encoding the TagRFP was first amplified with primers that removed the "A" nucleotide of the start codon. The PCR product was then cloned into a plasmid containing the GFP coding sequence and inserted directly downstream of the GFP's stop codon "TAA". A DNA fragment that contains a 3.3 kb *mec-3* promoter and the genomic coding region from exon 1 to exon 3 (with an extra nucleotide inserted downstream to exon 3) were cloned and inserted upstream of GFP (Fig 5A). This reporter was then injected into the wild-type animals and the *mbl-1(-)* mutants to form extrachromosomal arrays. Three "YGCU(U/G)Y" sites on the reporter construct were deleted individually or in combination to create the mutant reporters, which were injected into the wild-type animals to examine the effects of the *cis*-regulatory sites in regulating *mec-3* alternative splicing. To prevent nonsense-mediated decay, all experiments were done in the *smg-1 (cc546ts)* background. Strains were maintained at 15°C and tested at 25°C by growing the animals from eggs to adults in a 25°C incubator.

## Phenotype scoring and statistical analysis

Fluorescent imaging was performed on a Leica DMi8 inverted microscope with a Leica K5 monochrome camera, and images were analyzed using the Leica Application Suite X (3.7.2.22383) software. Measurement of ALM-PN and PLM-AN lengths were made on day-one adults grown at 20°C; at least 20 animals were measured for each strain. PLM-AN growth defects were quantified by measuring the distance from the PLM-AN terminal to the vulva. The distance is positive when the anteriorly directed PLM-AN grows beyond the vulva and is negative when PLM-AN fails to reach the vulva. To compare the fluorescence intensity between strains, day-one adults from different strains were prepared simultaneously and imaged using the same settings (e.g., 500ms exposure).

For statistical analyses, we utilized one-way ANOVA followed by Dunnett's multiple comparisons test of mutants with the wild-type animals or Tukey's Honestly Significant Difference (HSD) test for all pairwise comparisons using GraphPad Prism 9 or 10. Student's *t*-test was used to compare only one pair of samples. Quantitative data were plotted as mean ± SD, and adjusted $p$ values < 0.05 and 0.01 were indicated by one and two asterisks, respectively.

## Supporting information

**S1 Fig. Mutations in *mbl-1* suppress ectopic neurite growth induced by *klp-7(-)* mutants.** (A) ALM-PN length in the indicated strains. *klp-7(tm2143)* is a deletion allele, while *klp-7*

*(u1015; E95K)* is a loss-of-function allele we previously isolated. Double asterisks indicate $p < 0.01$ in a *t*-test comparing the indicated two strains. (B) The expression of a fosmid reporter *mbl-1::EGFP* in the TRNs based on the colocalization with the TRN marker *mec-17p:: TagRFP*. Enlarged images show the ALM and PLM cell bodies. Scale bar = 20 μm.
(TIF)

**S2 Fig. Splicing regulators EXC-7 and MEC-8 do not regulate neurite growth in TRNs.** (A) Protein domain structure of EXC-7 and MEC-8; RRM means RNA recognition motif. Gene structure of *exc-7* and *mec-8* and the molecular changes of their mutations. *unk71* and *unk73* are deletion alleles generated by CRISPR/Cas9-mediated gene editing in this study. (B) ALM-PN length in *mec-7(u278)* animals that also carry mutations in various splicing regulators. (C) Quantification of PLM-AN length by the distance from the vulva to the PLM-AN terminus. Positive values mean that PLM-AN grew beyond the vulva, while negative values mean PLM-AN failed to reach the vulva. In (B-C), double asterisks indicate $p < 0.01$ in a post-ANOVA Tukey's HSD test. "n.s." means no statistical significance. (D) Representative images of *exc-7* and *mec-8* mutants showing normal TRN morphologies.
(TIF)

**S3 Fig. MBL-1 promotes the splicing of *ptl-1* long isoforms.** (A) Exon structure of the *ptl-1* gene and the domain structures of the PTL-1 protein isoforms. The PE-rich domain is a region that contains many proline-glutamic acid motifs and is mostly coded by exon 1 and exon 2. The *unk137* allele (created in this study) has a stop codon (in red) inserted into the exon 2 of the gene. (B) RT-PCR results using isoform-specific primers. To detect the long isoforms (*ptl-1a*, *b*, and *c*), a pair of primers that recognize the exon 1-exon 2 junction and the exon 2–3 junction, respectively, was used. To detect all isoforms, primers that recognize the exon 3–4 junction and the exon 5-exon 6 junction, respectively, were used. (C) ALM-PN lengths of *mec-7(u278)* animals with a *ptl-1(unk137)* mutation that inactivated all long isoforms and of *mec-7(u278) mbl-1(tm1563)* animals with the overexpression of the *ptl-1a* or *ptl-1b* cDNAs or both from a TRN-specific *mec-17* promoter. "n.s." means no statistical significance.
(TIF)

**S4 Fig. MEC-3 splicing variants show distinct functions in activating TRN genes.** (A) Gene structure of *mec-3*. The *u184* allele serves as a *mec-3(-)* mutant, while *syb8382* and *unk206* alleles eliminate the *a* and *d* isoforms, respectively. The isoform-specific detection primers used in RT-PCR are indicated by dashed arrows. (B) RT-PCR results of the specific isoform in *mec-3* mutants. (C) Fluorescent intensity of the promoter reporter *mec-17p::TagRFP* in PLM neurons of various strains. For *mec-3(syb8382)* animals, only the fluorescence of cells that show clear RFP expression were quantified. No statistically significant difference (n.s.) was found between the wild-type and *mec-3(unk206)* animals. Double asterisks indicate $p < 0.01$ in a post-ANOVA Tukey's test. (D) Fluorescent intensity of the promoter reporter *mec-3p:: GFP* in PLM neurons of wild-type, *mec-3(unk206)*, and *mec-3(syb8382)* animals. For *mec-3 (syb8382)*, only the fluorescence of cells that show clear GFP expression were quantified. (E) RT-qPCR results of *mec-7*, *mec-12*, and *mec-17* mRNA levels in wild-type and *mbl-1(tm1563)* animals presented as Log2(fold change). (F) Representative images of *mec-17p::TagRFP* expression in ALM neurons in wild-type, *mbl-1(-)*, *mbl-1(-); mec-3(unk206)*, *mec-7(u278)*, *mec-7(u278) mbl-1(-)*, and *mec-7(u278) mbl-1(-); mec-3(unk206)* animals. (G-H) smFISH signals against the TRN genes *mec-7* and *mec-17* in the PLM neurons of various strains. *mec-3(a)* is the *unk206* allele and *mec-3(d)* is the *syb8382* allele. Double asterisks indicate $p < 0.01$ in Tukey's HSD tests in comparison with the wild-type animals or between specific pairs.
(TIF)

**S5 Fig. MEC-3 regulates *mbl-1* transcription in the TRNs.** (A) Expression of two promoter reporters for different *mbl-1* isoforms, *mbl-1p1::GFP* and *mbl-1p2::GFP* (also see Fig 1E), in the ALM, AVM, and PLM neurons. The wild-type and *mec-3(unk206)* animals showed the TRN expression indicated by the arrows, while the *mec-3(u184)* and *mec-3(syb8382)* mutants showed the lack of expression indicated by the dashed circles. Scale bar = 20 μm. (B) The expression of the *mbl-1* fosmid reporter *wgIs664[mbl-1::TY1::EGFP]*, which labels all isoforms of *mbl-1*, was lost in TRNs (dashed circle) in *mec-3(u184)* and *mec-3(syb8382)* mutants. (TIF)

**S6 Fig. Correcting *mec-3* splicing defects or overexpressing tubulin genes does not rescue the neurite growth defects in *mbl-1(-)* mutants.** (A) PLM-AN length measured by the distance from the PLM-AN terminus to the vulva in *mbl-1(tm1563)* mutants with overexpression of *mec-3a* cDNA or the non-spliceable *mec-3(unk206)* allele. (B) ALM-PN length in *mec-7 (u278) mbl-1(-)* animals with overexpression of *mec-3a* cDNA or the *mec-3(unk206)* allele. (C) PLM-AN length in *mbl-1(-)* mutants with overexpression of the tubulins *mec-7(+)* or *mec-12 (+)* from a *mec-17* promoter. (D) ALM-PN length in *mec-7(u278) mbl-1(-)* animals with overexpression of the specific cDNA isoforms of five genes whose splicing were affected by *mbl-1 (-)* mutation, as well as in *mec-7(u278) mbl-1(-)* animals with the overexpression of *mec-7(+)* or *mec-7(C303Y)*, which is the mutation found in *mec-7(u278)*, or *mec-12(+)*. (E) PLM-AN length in *mbl-1(-); mec-3(unk206)* mutants with *mec-7(+)* overexpression from a *mec-17* promoter. (F) ALM-PN length in *mec-7(u278) mbl-1(-); mec-3(unk206)* mutants with the overexpression of *mec-7(+)* or *mec-7(C303Y)*. (TIF)

**S7 Fig. Deletion of DLK-1 pathway genes does not rescue PLM-AN growth in *mbl-1(-)* mutants.** (A) The genomic region that is deleted in the *unk61* allele created in this study. (B) PLM-AN length measured by the distance from the PLM-AN terminus to the vulva in *mbl-1 (tm1563) dlk-1(ju476)* and *mbl-1(-) uev-3(unk61)* double mutants; *ju476* is a 5-bp insertion allele that caused frameshift in *dlk-1*. (C) The number of EBP-2::GFP comets in *mbl-1(-) mec-3 (unk206)* and *mbl-1(-) dlk-1(ju476)* in a 60 μm region from the PLM cell body within a one-minute recording. Single asterisks indicate $p < 0.05$ in a post-ANOVA Dunnett's test. (D) ALM-PN length in *mec-7(u278) mbl-1(-); mec-3(unk206)* and *mec-7(u278) mbl-1(-); dlk-1 (ju476)* triple mutants and *mec-7(u278) mbl-1(-); dlk-1(ju476); mec-3(unk206)* quadruple mutants. "n.s." means no statistical significance in a post-ANOVA Tukey's test. (TIF)

**S8 Fig. MBL-1 regulates the development of PVD dendrites.** Representative images of PVD morphologies anterior to the cell body in various strains. Arrows in the wild-type image pointed to the secondary, tertiary, and quaternary level dendrites. Asterisks in the *mbl-1(-)* mutant image indicated the absence of quaternary dendrites on these tertiary dendrites. Arrow heads in the *mbl-1(-); mec-3(unk206)* mutants indicate the restoration of the quaternary dendrites. (B) Quantification of the number of secondary, tertiary, and quaternary dendrites in a region anterior to the PVD cell body but ≤ 200 μm away from the cell body. *mec-3(-)*, *mec-3 (a)*, and *mec-3(d)* indicate the use of *u184*, *unk206*, and *syb8382* alleles, respectively. Double asterisks indicate $p < 0.01$ in Tukey's HSD tests in comparison with the wild-type animals or between specific pairs. (TIF)

**S1 Table. Significantly upregulated and downregulated exon-exon junctions in the *mbl-1 (tm1563)* mutants compared to the wild-type animals.** (XLSX)

**S2 Table. Significantly upregulated and downregulated exon-exon junctions in the *mec-7 (u278) mbl-1(tm1563)* double mutants compared to the *mec-7(u278)* single mutants.**
(XLSX)

**S3 Table. Results of the functional tests of MBL-1 target genes through the overexpression of downregulated isoforms or the inactivation of upregulated isoforms.**
(XLSX)

**S4 Table. Genes differentially expressed between the *mbl-1(tm1563)* mutants and the wild-type animals.** Genes with a *p* value smaller than 0.05 were listed, regardless of the adjusted *p* values.
(XLSX)

**S5 Table. Genes differentially expressed between the *mec-7(u278) mbl-1(tm1563)* double mutants and the *mec-7(u278)* single mutants.** Genes with a *p* value smaller than 0.05 were listed, regardless of the adjusted *p* values.
(XLSX)

**S6 Table. Primers, CRISPR targets, constructs, and strains used in this study.**
(XLSX)

**S1 Appendix. Raw numeric data for all graphs in this paper.**
(XLSX)

## Acknowledgments

We thank the Caenorhabditis Genetics Center, which is funded by the National Institutes of Health (NIH) Office of Research Infrastructure Programs (P40 OD010440), and the National BioResource Project (NBRP), which is funded by the Japanese government, for providing strains.

## Author Contributions

**Conceptualization:** Chaogu Zheng.

**Data curation:** Ho Ming Terence Lee, Hui Yuan Lim, Haoming He, Chaogu Zheng.

**Formal analysis:** Ho Ming Terence Lee, Hui Yuan Lim, Haoming He, Chun Yin Lau, Chaogu Zheng.

**Funding acquisition:** Chaogu Zheng.

**Investigation:** Ho Ming Terence Lee, Chaogu Zheng.

**Methodology:** Ho Ming Terence Lee, Hui Yuan Lim, Haoming He, Chun Yin Lau, Chaogu Zheng.

**Project administration:** Chaogu Zheng.

**Resources:** Ho Ming Terence Lee.

**Software:** Chun Yin Lau.

**Supervision:** Chaogu Zheng.

**Validation:** Ho Ming Terence Lee, Chaogu Zheng.

**Visualization:** Ho Ming Terence Lee, Hui Yuan Lim, Haoming He, Chun Yin Lau, Chaogu Zheng.

**Writing – original draft:** Chaogu Zheng.

**Writing – review & editing:** Ho Ming Terence Lee, Chaogu Zheng.

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
