## [Decision Letter · Decision Letter 0]

12 Jul 2024

Dear Dr Zheng,

Thank you very much for submitting your Research Article entitled 'MBL-1/Muscleblind regulates neuronal differentiation and controls the splicing of a terminal selector in Caenorhabditis elegans' to PLOS Genetics.

The manuscript was fully evaluated at the editorial level and by independent peer reviewers. The reviewers appreciated the attention to an important topic but identified some concerns that we ask you address in a revised manuscript. 

We therefore ask you to modify the manuscript according to the review recommendations. Your revisions should address the specific points made by each reviewer.

1) Provide a detailed list of your responses to each of the review comments and a description of the changes you have made in the manuscript.

To resubmit, log into your Editorial Manager account and select the option 'Revise Submission' in the 'Submissions Needing Revision' folder.

Yours sincerely,

Anne C. Hart

Academic Editor

PLOS Genetics

Pablo Wappner

Section Editor

PLOS Genetics

Reviewer's Responses to Questions

**Comments to the Authors:**

Reviewer #1: This manuscript by Lee et al. reports a nice and thorough investigation into the molecular consequences of mbl-1 loss on a single neuron type (touch neurons). It presents important points about the relationship between RNA processing and neuronal differentiation/fate. The interpretations are well-founded, and appropriate controls are present. A few changes will make the manuscript suitable for publication:

1. RNA Seq is data is generated but does not appear to be public at this time (on GEO, SRA, or similar databases).

2. On page 8, the claim is made that mbl-1(-) causes gene expression changes in mec-12, mec-7, and mec-17. However, the data presented in Figure 4 show that the changes in mec-12 and mec-17 are not significant. Therefore the claims on page 8 should be removed, as claims about statistically-insignificant gene expression changes should not be reported as gene expression changes.

3. Page 4: a brief sentence explaining what EBP-2::GFP imaging is measuring (e.g. what is EBP-2? Where does it localize?) This would help non-specialists such as myself.

Reviewer #2: In this manuscript, Lee and colleagues identified mbl-1 as a genetic suppressor of mec-7(u278) using the ectopic neurite outgrowth as the readout. mbl-1(lf) also suppresses the ectopic neurite formation in klp-7(lf) mutants, which is consistent with the results reported by the Ghosh-Roy group (PMID: 37603562). Since mec-7(u278) causes hyperstable microtubules, the authors examined the MT stability and dynamics and found that mbl-1(lf) animals showed reduced MT stability and increased MT dynamics when compared to the WT. They performed RNAseq and splicing junction analysis and identified many alternative splicing events controlled by MBL-1. Importantly, the splicing of mec-3, a terminal selector critical for the cell fate determination of the TRN and PVD neurons, is dramatically altered by mbl-1(lf). The level of mec-3(a) mRNA is much reduced while that of mec-3(d) is increased in mbl-1(lf). MEC-3(a) shows higher activity to promote the expression of the mec-3 downstream genes, including mec-7, mec-12, and mec-17. Using the CRISPR technology, the authors generated mec-3(syb8382, delta exon 2) and mec-3(unk206, delta intron 1 and intron 2) and showed that correcting the mec-3 alternative splicing defects could partially restore the expression of mec-7 and mec-17. Why is the suppression not 100%? One possibility is that MBL-1 can promote the stability of the mRNAs of the mec-3 target genes, which is reported by the Ghosh-Roy group. The authors further generated a mec-3 splicing reporter and identified three “YGCU(U/G)Y” motifs which can be recognized by MBL-1. Through another round of genetic suppressor screen, they found that the ectopic ALM-PN could form again in mec-7(u278); mbl-1(lf) when dlk-1/pmk-3/cebp-1 is inactivated. Finally, they showed that mbl-1(lf) mutants showed reduced 4o dendrite formation in the PVD neurons, which is largely due to the mec-3 splicing defects (consistent with the findings reported by the Shen lab; PMID: 37729192).

Overall, this study is well-designed and the quality of the data is high. Most of the conclusions are well-supported by the results. The manuscript is well-written and easy to understand. In this independent study, the authors confirmed some of the results reported by the Ghosh-Roy group and the Shen group, and brought some new insights for the mechanisms of how MBL-1 regulates neuronal cell fate determination and morphogenesis. Thus, it deserves to be published in PLoS Genetics. I only have some minor suggestions.

1. In Fig. 2, the authors showed that mbl-1(lf) caused instability of MT. Can this defect be rescued by paclitaxel treatment?

2. Any explanation why there are more mitochondria in the proximal part of PLM-AN? It seems to me that the number of mitochondria in the entire PLM-AN is increased in mbl-1(lf) when compared to the WT. Is that true?

3. I am curious whether the defects of MT stability and polarity in mbl-1(lf) can be suppressed by mec-3(unk206) and dlk-1(lf).

4. It would be valuable to compare the transcriptomes of mbl-1(lf) and mbl-1(lf); mec-3(unk206) and analyze the splicing junctions.

5. For Fig. 4B, please clearly indicate that the experiments were performed in the mec-3(lf) genetic background in this panel (although this information can be found in the figure legend). And explain why there is no much difference in the PLM neurons for the two reporters?

6. For Fig. 4C, also explain why the expression of mec-3p::gfp is reduced in mec-3(d) mutants, while that of mec-17p::tagRFP is not? How many animals were quantified for Fig. 4B and 4C? The information of the sample size should be indicated in the figure/figure legend/somewhere else.

7. For Fig. 4F, it’s better to indicate the results as: mec-7(u278); mbl-1(-)/mec-7(u278); and mbl-1(-)/WT on top of the panels.

8. For Fig. 4I, any significant difference between mbl-1(-) and mbl-1(-); mec-3(a)? Even if it is NS, it should be indicated.

9. For Fig. 5, the site 1 is clearly in the exon 2. Why did the authors claim that all the three sites are downstream of the included exon?

10. For Fig. 6, since mbl-1(lf) alone shows PLM-AN outgrowth defect, did the authors ever examine whether loss of dlk-1 and the downstream genes can suppress the above-mentioned defect?

11. The authors proposed that MBL-1 functions upstream of the DLK-1 pathway. The only result supporting this conclusion is the genetic suppression. I am not totally convinced as we may still see the suppression if they act in redundant genetic pathways. The ALM-PN outgrowth phenotype is just the readout of the overall neurite outgrowth activity of the ALM neuron.

12. For Fig. 7A, it is better if the order of the images is consistent with Fig. 7B.

13. For Fig. S1, it is nice to see the results using the mbl-1 fosmid-based reporter. In Fig. S5, the authors analyzed the two mbl-1 promoter-gfp fusion reporters and concluded that mec-3 positively regulates the transcription of mbl-1. Is the MBL-1 protein expression level affected by loss of mec-3? This can be done using the fosmid-based reporter.

14. The authors stated: the mec-3a was required for the expression of likely all isoforms of mbl-1 and mec-3d alone was not sufficient to activate mbl-1. The two transcriptional reporters do not report the transcription of the isoform d and e. Thus, it is not proper to say “all isoforms”.

15. For Fig. S8B, the dendrite formation phenotypes are quantified in different regions for different genotypes, which is confusing. As mbl-1(lf) mutants show reduced branching in the distal part, but not the proximal part, of the PVD neurons, it is better to show the number of dendritic branches in the 200 um region anterior to the vulva for all the strains. mec-3(d) was incorrectly labeled as mec-3(b). m3a should be labeled as mec-3(a).

16. For Table S6, the authors should list all the strains used in this study in the order of the figures, and clearly indicate the figures in which the strains are used. At least some of the strains used in Figure 7 are not included in the current version.

Reviewer #3: In this manuscript, a genetic screen by Lee et al. unravels the role of the splicing regulator mbl-1 in promoting neurite growth, consistent with the findings of Puri et al. (2023). The authors of this manuscript also demonstrate the role of mbl-1 in regulating the splicing of mec-3, the terminal selector for the touch receptor neuron (TRN) fate, which supports the findings of Xie et al. (2023). Overall, this manuscript is a valuable contribution to existing literature and the authors have done well to support their claims with solid data. In particular, the experiments demonstrating the role of mbl-1 in regulating mec-3 splicing are elegant and compelling.

The authors begin the manuscript with a genetic screen to reveal neurite outgrowth regulators, in which they find mbl-1. They demonstrate convincingly that mbl-1 regulates neurite outgrowth. In attempting to find targets of mbl-1, they perform RNA-seq, in which they discover that mbl-1 regulates the splicing of seven TRN-expressed genes. Among these genes, they focus on mec-3 because it is the only gene in which, upon overexpression of the mec-3d isoform upregulated in mbl-1(-), they find a suppression of neurite outgrowth which is a similar phenotype to mbl-1(-), albeit milder. They then proceed to present compelling evidence that mbl-1 regulates mec-3 splicing. Overall, the authors did well to support their claims that (1) mbl-1 regulates neurite outgrowth and (2) mbl-1 regulates mec-3 splicing.

Major comments:

In Fig. 3C. Since much of the data contained in the paper rest on the differential isoform usage of mec-3 in mbl-1 mutant animals, we suggest performing a qPCR to measure the expression levels of mec-3a and mec-3d in WT, mbl-1, mec-7 and mec-7;mbl-1 mutants (See as example Fig6F in this manuscript and Fig6C in Puri et al.). This would corroborate the RT-PCR data. In addition, the authors could also measure the overall levels of mec-3 in all the conditions mentioned above using qPCR. While less critical, this experiment could help understand whether mbl-1 is specifically controlling splicing of mec-3 or also its overall levels. The authors also need to show quantification and statistical analysis while comparing RT-PCR band intensities in Fig 3C.

In mbl-1(-), the expression of terminal identity genes such as mec-7 and mec-17 is reduced (Fig4G, 4H, 4I). This could be due to the aberrant expression of mec-3d in mbl-1 mutants, but the authors show removing mec-3d is not sufficient to restore control levels of mec-7 and mec-17 (Fig 4h and 4I). The authors suggest mbl-1 could target both mec-3 (as shown in this manuscript) and mec-7 (as shown by Puri et al.). To corroborate this interesting model the authors should test if they can rescue the ALM and PLM neurite phenotype in an mbl-1 background where the mec-3d isoform has been removed (mec-3(A)) and mec-7 expression has been restored. The ideal experiment would be to find if the same YGCU(U/G)Y motif is also present in mec-7 and mutate it so as to abrogate the possible direct regulation between mbl-1 and mec-7. This approach was very successful for the authors in this manuscript (See Fig5). If this is not feasible then the authors should try again to over-express mec-7 in a mbl-1;mec-3(a) background to test if this is sufficient to fully rescue the ALM-PN phenotype (e.g mec-7(u278) vs mbl-1;mec-3(a);mec-17p:mec-7 or the PLM-AN phenotype.

The authors nicely show that overexpressing both mec-3d and dlk-1a has an additive effect in suppressing mec-7(u278) extended ALM-PN neurite. They should also test if, in an mec-7(u278);mbl-1 background, removing mec-3d isoform (using mec-3(a) allele) together with removing dlk-1 (or any other member of the MAPK pathway) has a stronger effect than removing dlk-1 alone. This should be done by testing (as an example) the ALM-PN neurite’s length in mec-7(u278);mbl-1 vs mec-7(u278);mbl-1;mec-3(d) vs mec-7(u278);mbl-1;dlk-1 vs mec-7(u278);mbl-1;mec-3(d);dlk-1.

Minor comments:

Please indicate the allele for mec-17p::TagRFP reporter in Fig. 4D. In the methods section, two mec-17p::TagRFP reporter alleles are mentioned, uIs115 and uIs134, but only uIs115 is cited in the main text.

The authors should provide more details about their RNAseq experiments. It would be useful to show some standard quality control plots like a PCA plot with all the samples.

The authors should address the fact that mec-7(u278) seems to have an effect on the expression of several genes that are also regulated by mbl-1 (for example WT vs mec-7(u278) on 1) mec-3a and mec-3d(perhaps) in Fig3C 2) ptl-1a,b,c and ptl-1a-d in FigS3 3) mec-17p:TagRFP in Fig4G. A possible regulation of these genes by changes in mec-7 could further complicate the model the authors are presenting. Considering that mbl-1 affect the expression of mec-7 this possibility should be at least addressed.

Puri et al. (2023) shows that the mbl-1(-) allele tm1563 reduces anterior and posterior touch sensitivity. The author’s data in Fig 4J for anterior touch is similar to Puri et al., although they used a touch response index to measure sensitivity (versus number of responses for this manuscript). However, while Puri et al. sees an effect for posterior touch, this manuscript does not. The authors should mention this in this manuscript.

Authors mention that skipping of exon 2 by shorter mec-3 isoforms leads to truncation of the LIM domain. Looking at the genome browser in Wormbase however, it does not seem that exon 2 contributes to the LIM domain. In Fig4A, the authors should put the schematic of LIM domain motifs directly below the schematic of the exon structure of mec-3 to clarify the correspondence between the two.

To improve clarity, the authors can show locations of the primers they used in figures illustrating gene loci/exon structures. For example in Fig3B for gene egl-8 it’s unclear where the primers sit on the gene structure. Is the first exon, exon1? Which exon is the last one represented? This could be made clearer.

Line-by-line comments:

Line16 - regulates

Line22 - dash missing in MLB-1

Line25 - is instead of are

Figure 1F - Very hard to tell the difference in GFP+RFP between the two channels. Would be better to split and add merge like FigS1B

Line137 - dash missing in mbl-1

Fig 2B - need to add Y label title to make it clearer

Line170 - It could helpful (especially to people outside the field) if the authors add one sentence explaining why they are using mitochondria as an example of MT-mediated cargo transport

Line206 - Authors don’t know for sure that MBL-1 function is unique. A more accurate statement would be “ability to control neural growth not general to all splicing factors”

Line237 - The OE of ptl-1 isoforms and the effect in ALM-PN seems to miss from FigS3

Line303 - typo in “/-tubulin” ?

Line306 - Fig4D does not show results for mbl-1 (-) allele

Line 384 - missing dash in mbl-1

Line 418 - Typo “loss-offunction”

Line446-451- This paragraph could be misleading and should be changed. The fact that the authors did not find MAPK pathway genes as candidates in their RNAseq is not sufficient evidence to suggest mbl-1 does not regulate genes in this pathway.

**Have all data underlying the figures and results presented in the manuscript been provided?**

Reviewer #1: **No: **RNA Seq is data is generated but does not appear to be public at this time (on GEO, SRA, or similar databases)

Reviewer #2: Yes

Reviewer #3: Yes

PLOS authors have the option to publish the peer review history of their article (what does this mean?). If published, this will include your full peer review and any attached files.

Reviewer #1: No

Reviewer #2: No

Reviewer #3: No

---

## [Decision Letter · Decision Letter 1]

9 Oct 2024

Dear Dr Zheng,

We are pleased to inform you that your manuscript entitled "MBL-1/Muscleblind regulates neuronal differentiation and controls the splicing of a terminal selector in Caenorhabditis elegans" has been editorially accepted for publication in PLOS Genetics. Congratulations!

Yours sincerely,

Anne C. Hart

Academic Editor

PLOS Genetics

Pablo Wappner

Section Editor

PLOS Genetics

Comments from the reviewers (if applicable):

Reviewer's Responses to Questions

**Comments to the Authors:**

Reviewer #1: My concerns have been fully addressed. My congratulations to the authors on a nice paper.

Reviewer #2: All my concerns have been properly addressed. I recommend public communication of this work.

Reviewer #3: This authors have done an excellent job in addressing reviewer's comments and the manuscript is now acceptable as is.

**Have all data underlying the figures and results presented in the manuscript been provided?**

Reviewer #1: Yes

Reviewer #2: Yes

Reviewer #3: None

PLOS authors have the option to publish the peer review history of their article (what does this mean?). If published, this will include your full peer review and any attached files.

Reviewer #1: No

Reviewer #2: No

Reviewer #3: No

**Data Deposition**

http://datadryad.org/submit?journalID=pgenetics&manu=PGENETICS-D-24-00462R1

**Press Queries**

---

## [Editor Report · Acceptance letter]

14 Oct 2024

PGENETICS-D-24-00462R1 

MBL-1/Muscleblind regulates neuronal differentiation and controls the splicing of a terminal selector in Caenorhabditis elegans 

Dear Dr Zheng, 

We are pleased to inform you that your manuscript entitled "MBL-1/Muscleblind regulates neuronal differentiation and controls the splicing of a terminal selector in Caenorhabditis elegans" has been formally accepted for publication in PLOS Genetics! Your manuscript is now with our production department and you will be notified of the publication date in due course.

With kind regards,

Zsofia Freund

PLOS Genetics

On behalf of:
